# CCN1 interacts with integrins to regulate intestinal stem cell proliferation and differentiation

Jong Hoon Won [1], Jacob S. Choi [1,2] & Joon-Il Jun [1✉]

Intestinal stem cells (ISCs) at the crypt base contribute to intestinal homeostasis through a balance between self-renewal and differentiation. However, the molecular mechanisms regulating this homeostatic balance remain elusive. Here we show that the matricellular protein CCN1/CYR61 coordinately regulates ISC proliferation and differentiation through distinct pathways emanating from CCN1 interaction with integrins $\alpha_v\beta_3/\alpha_v\beta_5$. Mice that delete *Ccn1* in *Lgr5* + ISCs or express mutant CCN1 unable to bind integrins $\alpha_v\beta_3/\alpha_v\beta_5$ exhibited exuberant ISC expansion and enhanced differentiation into secretory cells at the expense of absorptive enterocytes in the small intestine, leading to nutrient malabsorption. Analysis of crypt organoids revealed that through integrins $\alpha_v\beta_3/\alpha_v\beta_5$, CCN1 induces NF-κB-dependent *Jag1* expression to regulate Notch activation for differentiation and promotes Src-mediated YAP activation and *Dkk1* expression to control Wnt signaling for proliferation. Moreover, CCN1 and YAP amplify the activities of each other in a regulatory loop. These findings establish CCN1 as a niche factor in the intestinal crypts, providing insights into how matrix signaling exerts overarching control of ISC homeostasis.

[1] Department of Biochemistry and Molecular Genetics, College of Medicine, The University of Illinois at Chicago, 900 South Ashland Avenue, Chicago, IL 60607, USA. [2] Present address: Department of Medicine, Northwestern University, 676 North St. Clair street Arkes Suite 2330, Chicago, IL 60611, USA. ✉email: jjun7@uic.edu

The main function of the intestine is to efficiently absorb nutrients and water while enduring mechanical force, extreme pH variations, and constant exposure to the gut microbiota. To avoid the accumulation of damaged cells while achieving these functions, the intestine continuously sheds aged epithelial cells and rapidly regenerates the crypt-villus structure every 4–5 days[1]. The villus is a protrusion of epithelium toward the intestinal lumen to maximize the surface area for efficient nutrient absorption and contains a single columnar layer of fully differentiated post-mitotic epithelial cells. Actively dividing stem cells reside at the crypt, an epithelial invagination surrounding the villi, and proliferate for self-renewal or generate progenitor cells that move up to the transit-amplifying (TA) zone to rapidly divide before committing into epithelial cells of specific lineages to replenish the aged and damaged cells in the villi. To maintain homeostasis, the balance between self-renewal and differentiation of the intestinal stem cells (ISCs) needs to be fine-tuned and sustained.

Genetic lineage-tracing identified a stem cell population at the bottom of the crypts that expresses the leucine-rich repeat-containing G protein-coupled receptor 5 (LGR5) and is fully capable of long-term self-renewal and generation of both absorptive cells (enterocytes and M cells) and secretory cells (Paneth cells, goblet cells, enteroendocrine cells, and tuft cells)[1,2]. These stem cell activities are largely governed by the surrounding microenvironment[3], the so-called stem cell niche, which instructs ISCs to proliferate and/or differentiate through cellular signaling driven by several niche factors[4–8], including Wnt, Notch ligands, epidermal growth factor (EGF), and bone morphogenic protein (BMP). Especially, Wnt and Notch signaling are crucial determinants in regulating ISC proliferation and differentiation, respectively. The canonical Wnt activation, initiated by the binding of secreted Wnt ligands to the cognate receptors frizzled (FZD) and low-density lipoprotein receptor-related protein 5/6 (LRP5/6), leads to the nuclear accumulation of β-catenin and activation of T cell factor 4 (TCF4) target genes involved in ISC proliferation and maintenance[9]. Mice with systemic[10] or epithelial deletion of *Tcf4* (ref. [11]), epithelial deletion of β-catenin[12], or overexpression of a secreted Wnt antagonist Dickkopf1 (Dkk1)[13] exhibited rapid ablation of ISCs and subsequent failure of the intestinal epithelium. On the other hand, Notch signaling is critically involved in ISC fate decision to differentiate toward cells of the absorptive or secretory lineages. Upon Notch ligand-receptor interaction between neighboring cells, Notch signaling induces the expression of hairy and enhancer of split 1 (*Hes1*) to suppress atonal homolog 1 (*Atoh1*), a basic helix-loop-helix (bHLH) transcription factor that promotes the transcriptional program for

differentiation toward cells of the secretory lineage[14,15]. Accordingly, the conditional deletion of recombination signal binding protein for immunoglobulin kappa J region (*Rbp-j*), a transcription factor mediating Notch signaling, or administration of the γ-secretase inhibitor dibenzazepine (DBZ) that blocks Notch processing caused a massive conversion of proliferative crypt cells into goblet cells[16,17]. Despite the importance of these signaling pathways in ISC homeostasis, the overarching mechanisms for the coordinated regulation of Wnt and Notch signaling pathways remain elusive.

Cellular communication network factor 1 (CCN1, previously named CYR61) is a dynamically expressed, secreted matricellular protein essential for embryonic development[18,19]. In the adult, *Ccn1* expression and function are required for successful injury repair and regeneration in the skin[20,21], liver[22], heart[23], and gut[24], where CCN1 exerts diverse cellular activities, including cell migration, proliferation, apoptosis, senescence, and phagocytosis in a cell type-specific manner, largely through its direct binding to specific integrin receptors[25,26]. Importantly, CCN1 also regulates or is regulated by factors including Wnt[27], Notch ligands[22,28], and BMPs[29], suggesting its role in stem cell functions. Indeed, CCN1 induces endothelial differentiation of cultured hematopoietic stem cells[30] and restricts the expansion of neural stem cells in adult mouse brain[31]. Although CCN1 is critical for intestinal mucosal healing[24], the precise roles of CCN1 in ISCs and specific integrins involved are unknown.

Here we provide the evidence that CCN1 coordinately regulates ISC proliferation and differentiation during homeostasis, and *Ccn1* deletion in ISCs resulted in enhanced ISC proliferation and increased secretory cells at the expense of absorptive cells. Analysis of crypt organoids revealed that CCN1 regulates both Notch and Wnt signaling for differentiation and proliferation of ISCs, respectively, through distinct pathways emanating from the engagement of integrins $\alpha_v\beta_3/\alpha_v\beta_5$. Moreover, a CCN1-yes-associated protein (YAP) regulatory loop amplifies their activities in the crypts. Together, these findings identify CCN1 as a niche factor in the matrix that activates integrins $\alpha_v\beta_3/\alpha_v\beta_5$ signaling to coordinately regulate ISC proliferation and differentiation through distinct downstream pathways, thus expanding our understanding of the overarching signaling mechanisms contributing to the maintenance of intestinal homeostasis.

## Results

**Ccn1 deletion in Lgr5 + ISCs engendered nutrient malabsorption.** We examined *Ccn1* expression in the normal mouse small intestine to study its functions. Immunostaining of green

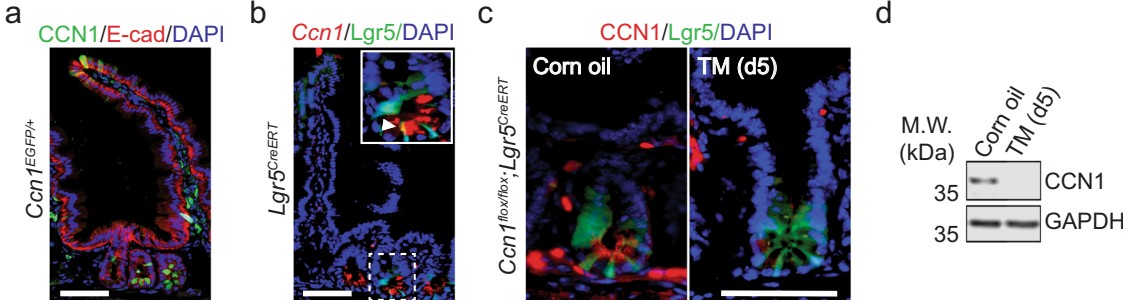

**Fig. 1 Ccn1 expression in the intestinal crypt. a** CCN1 expression (GFP) in the jejunum of *Ccn1^EGFP/+* mice. E-cadherin (E-Cad, red) marks epithelial cells. DAPI is counterstaining. Bar: 50 μm. **b** Fluorescence in situ hybridization of *Ccn1* in the jejunum of *Lgr5^CreERT* mice. Crypt (dotted box) is enlarged into a solid box. Arrowhead points *Ccn1* (red) in *Lgr5* + ISCs (GFP). Bar: 50 μm. **c** Immunofluorescence staining of CCN1 (red) in the crypt of *Ccn1^ΔLgr5* mice treated with corn oil or tamoxifen (TM, day 5). Bar: 50 μm. For **a**, **b**, and **c**, representative images are shown from three independent replicates. **d** Representative immunoblot on CCN1 expression in the jejunum of *Ccn1^ΔLgr5* mice with corn oil or TM (n = 3 each, d5). Glyceraldehyde-3-phosphate dehydrogenase (GAPDH) expression indicates equal loading.

fluorescent protein (GFP) in the jejunum of *Ccn1*:EGFP reporter mice (*Ccn1*[EGFP/+]) showed strong CCN1 expression in the lower base of the crypts where stem cells reside (Fig. 1a). Fluorescence in situ hybridization (FISH) also detected *Ccn1* mRNA signals from *Lgr5*-expressing (+) ISCs in *Lgr5*-EGFP-IRES-*Cre*ERT2 mice (*Lgr5*[EGFP-CreERT]; Fig. 1b). The secreted CCN1 protein was found along the luminal and basal side of the epithelium at the crypt base by immunohistochemistry, surrounding the *Lgr5* + ISCs and the neighboring Paneth cells, although its expression was also seen in a few epithelial cells and mesenchymal cells in the villi and lamina propria (Fig. 1c). A similar CCN1 expression pattern was observed in the crypts of the duodenum and ileum as well as the colon (Supplementary Fig. S1a, b). Thus, we postulated that CCN1 may play a role in *Lgr5* + ISCs in the crypts of the intestine. To test this possibility, we created *Ccn1* conditional knock-out mice (*Ccn1*[flox/flox]; *Lgr5*[EGFP-CreERT], hereafter *Ccn1*[ΔLgr5]) that delete *Ccn1* in *Lgr5* + ISCs upon tamoxifen (TM) treatment (Fig. 1c). Indeed, TM administration abolished CCN1 expression in the crypt base without affecting mesenchymal expression (Fig. 1c). Immunoblot analysis further confirmed that *Lgr5* + ISCs are responsible for CCN1 expression in the crypts (Fig. 1d).

*Ccn1*[ΔLgr5] mice treated with corn oil (vehicle) or TM showed similar bodyweight within the first 5 days, beyond which TM-treated mice ceased to gain weight, and by day 9 weight loss was evident, implying defects in nutrient uptake (Fig. 2a). *Ccn1*[flox/flox] mice with the same TM regimen did not show bodyweight loss, ruling out possible side effects of TM. Interestingly, the jejunum of *Ccn1*[ΔLgr5] mice exhibited structural changes by histology (H&E) on day 5-post TM treatment (Fig. 2b). The villus length was reduced by ~28% with increased blobs on the epithelium, whereas the crypt became slightly elongated by ~17% with more eosinophilic granules (Fig. 2b). Similar changes in villus length and crypt depth were also found in the duodenum and ileum of *Ccn1*[ΔLgr5] mice upon TM treatment (Supplementary Fig. S1d–g). To assess any effect on nutrient uptake and metabolism, *Ccn1*[ΔLgr5] mice on day 10-post vehicle or TM treatment were orally gavaged with sucrose and blood glucose was measured thereafter (Fig. 2c). Blood glucose level reached its peak within 15 min and was maintained for up to 60 min before its decline in vehicle-treated mice. By contrast, blood glucose was 50% lower in *Ccn1*[ΔLgr5] mice with TM at 15 min and further declined to background level by 30 min, indicating impaired sugar absorption in mice with *Ccn1* deletion in *Lgr5* + ISCs. Consistently, immunoblot analysis showed a greatly reduced expression of sucrase that hydrolyzes sucrose into glucose and fructose and Na$^+$K$^+$ ATPase β1 which is critical for generating Na$^+$ gradient for glucose uptake (Fig. 2d). qPCR analysis confirmed that enterocyte-specific genes involved in glucose uptake and transport (*Atp1a1*, *Atpab1*, *Slc2a1*, *Slc2a2*, and *Slc5a1*) were downregulated in TM-treated *Ccn1*[ΔLgr5] mice (Fig. 2e). In addition, the paracellular transport in the epithelium was also altered, as orally gavaged fluorescein isothiocyanate (FITC)-dextran was detected at significantly increased amounts in the bloodstream of TM-treated *Ccn1*[ΔLgr5] mice (Fig. 2f) and consistent downregulation of genes involved in tight junction formation (*Claudin-1*, *−2*, *−3*, and *−8*, and *Zo-2*) was observed in qPCR analysis (Fig. 2g). Thus, *Ccn1* is expressed in the intestinal crypts and its deletion in *Lgr5* + ISCs engenders architectural and functional alteration of the intestinal epithelium, resulting in nutrition malabsorption.

**Ccn1 deletion in Lgr5 + ISCs affected ISC homeostasis.** We then monitored the cellular composition in the jejunum of *Ccn1*[ΔLgr5] mice. Surprisingly, expansion of *Lgr5* + ISCs in the crypts, assessed by GFP expression (Fig. 3a, b), became evident by

day 5 of TM treatment and this was confirmed by a marked increase in the expression of olfactomedin 4 (Olfm4), another stem cell marker (Fig. 3a, b). The expanded *Lgr5* + ISCs still maintained their topological position at the crypt base. However, there was no evidence of hyperplasticity even by day 28 of TM treatment (Supplementary Fig. S2). qPCR analysis on the jejunum of *Ccn1*[ΔLgr5] mice with TM also showed increased expression of ISC signature genes, such as *Olfm4*, *Ascl2*, and *Sox9* (Fig. 3c). Moreover, ISC expansion was accompanied by increased Ki67+ proliferating cells in the middle region of the crypts, presumably the TA zone (Fig. 3a). Crypts from *Ccn1*[ΔLgr5] mice with TM formed twice as many colonies in 3D cultures as those from vehicle-treated mice when seeded equally, demonstrating the presence of more functional ISCs (Fig. 3d). We also assessed the composition of fully differentiated epithelial cells. Judging from enzymatic staining of alkaline phosphatase (AP), TM-treated *Ccn1*[ΔLgr5] mice showed a marked reduction in enterocytes throughout the entire villi (Fig. 3e), which might explain the structural changes in the villi, poor nutrient absorption, and reduced sugar metabolism in these mice (Fig. 2). By contrast, secretory lineage cells were increased by immunofluorescence staining and qPCR analysis of markers for Paneth cells (lysozyme 1, Lyz1), goblet cells (mucin 2, Muc2), and enteroendocrine cells (chromogranin A, ChgA) (Fig. 3e, f).

CCN1 functions in epithelial cells are mediated through integrins α$_v$β$_3$/α$_v$β$_5$ (ref. [22]) and we found the expression of integrins α$_v$ in the crypts (Supplementary Fig. S1a). Remarkably, *Ccn1*[D125A/D125A] knock-in mice (crossed with *Lgr5*[EGFP-CreERT] mice to mark the *Lgr5* + ISCs), expressing a mutant CCN1 with a single amino acid change (CCN1-D125A) that is unable to bind integrins α$_v$β$_3$/α$_v$β$_5$ (ref. [20]), also exhibited increased *Lgr5* + ISCs, Paneth cells (Lyz1), and goblet cells (Muc2), but a reduction in enterocytes in the jejunum by immunofluorescence staining, AP staining, and qPCR analysis (Fig. 3g, h), phenocopying *Ccn1*[ΔLgr5] mice with TM treatment. Together, these results indicate that *Ccn1* plays a role in both ISC proliferation and differentiation during homeostasis, most likely through integrins α$_v$β$_3$/α$_v$β$_5$.

**Curtailed Notch and enhanced Wnt signaling with Ccn1 deletion.** The compound phenotypes of *Ccn1*[ΔLgr5] mice with TM suggest the involvement of multiple signaling pathways, among which Notch and Wnt signaling are known to regulate ISC fate decision and proliferation, respectively[5]. The presence of the Notch1 intracellular domain (N1-ICD) indicated active Notch signaling in the crypts of vehicle-treated *Ccn1*[ΔLgr5] mice, which was greatly diminished with TM-mediated *Ccn1* deletion by immunofluorescence staining and immunoblotting (Fig. 4a, b). However, ICD of another Notch paralog Notch2 was unaffected (Fig. 4b). Despite their critical involvement in Notch activation of ISCs[32], the expression of Notch ligand Delta-like 1 (DLL1) or DLL4 was not affected (Fig. 4b). Instead, another ligand Jagged 1 (Jag1) was greatly reduced in the jejunum of *Ccn1*[ΔLgr5] mice with TM (Fig. 4b), suggesting that Jag1 deficiency may cause the reduced Notch activation. Notch ligand-receptor interaction is regulated by Notch receptor glycosylation[14,33]. Especially, the Fringe family (Manic, Lunatic, and Radical) of N-acetylglucosaminyltransferases glycosylate the Notch receptors, thereby potentiating DLL-induced signal but diminishing responsiveness to Jag[34,35]. Interestingly, the crypts of *Ccn1*[ΔLgr5] mice with TM exhibited a marked reduction in the expression of Manic Fringe (MFNG) (Supplementary Fig. S3a), which might alter the glycosylation status of Notch1 receptor and hence Notch ligand preference from DLLs to Jag1 (ref. [36]). Consequently, reduced Notch activation

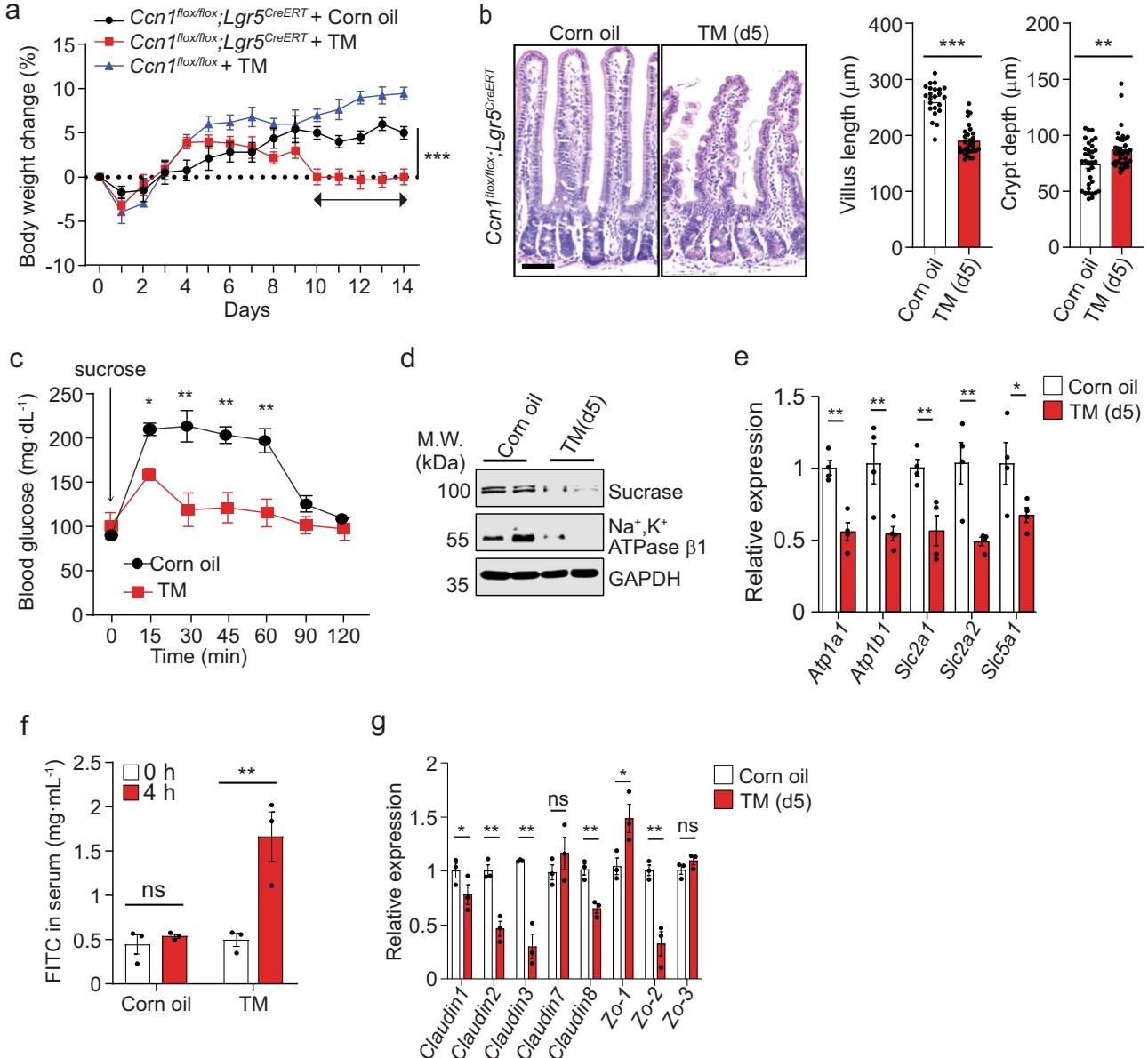

**Fig. 2 Changes in intestinal function with *Ccn1* deletion in *Lgr5* + ISCs. a** Bodyweight curve of $Ccn1^{\Delta Lgr5}$ mice with corn oil or TM ($n = 5$ each) for 14 days. $Ccn1^{flox/flox}$ mice ($n = 5$) were used as TM control. Data represent mean ± SEM. One-sided *t*-test. ***$p < 0.001$ (day 10–14). **b** H&E staining (*left*) of $Ccn1^{\Delta Lgr5}$ mice jejunum (day 5, $n = 3$ per group). Bar: 50 µm. The quantification of villus and crypt length (*right*) from 6 to 10 randomly selected fields per section using ImageJ. Data represent mean ± SEM. One-sided *t*-test. **$p < 0.01$; ***$p < 0.001$. **c** Blood glucose levels of $Ccn1^{\Delta Lgr5}$ mice at day 10-post corn oil or TM ($n = 3$ each) after sucrose oral-gavage (3 g per kg). Assays were performed in duplicate and data represent mean ± SEM. One-sided *t*-test. *$p < 0.05$; **$p < 0.01$. **d** Immunoblot analysis of sucrase and Na⁺, K⁺ ATPase β1. GAPDH is a loading control. Images are representative of three independent replicates. **e** qPCR analysis of enterocyte-specific genes (*Atpa1a, Atp1b1, Slc2a1, slc2a2, and Slc5a1*). Data represent the mean ± SEM of four biological replicates. One-sided *t*-test. *$p < 0.05$, **$p < 0.01$. **f** Paracellular transport of oral-gavaged FITC-dextran (MW 4000, 60 mg per kg; 4 h) in $Ccn1^{\Delta Lgr5}$ mice with corn oil or TM (day 10, $n = 3$ each). Blood was collected from an intraorbital sinus and fluorescence intensity in the serum was measured (excitation, 485 nm; emission, 520 nm) in triplicates. Data represent mean ± SEM. One-sided *t*-test. **$p < 0.01$; ns not significant. **g** qPCR analysis of tight junction-associated genes (*claudin-1, -2, -3, -7, and -8, Zo-1, −2, and −3*). Data represent the mean ± SEM of three biological replicates. One-sided *t*-test. *$p < 0.05$; **$p < 0.01$; ns not significant. In **c**, **e**, **f**, and **g**, all data were acquired from at least three independent assays. Source data and the exact *p*-values are provided in a Source Data file.

in $Ccn1^{\Delta Lgr5}$ mice with TM led to decreased expression of Notch target genes, including *Hes1*, *Hey1*, and *HeyL* (Fig. 4c and Supplementary Fig. S4a), resulting in derepression of *Atoh1* (Fig. 4c), which is consistent with the preferential secretory cell differentiation observed in $Ccn1^{\Delta Lgr5}$ mice with TM (Fig. 3e). By contrast, Wnt signaling was enhanced upon *Ccn1* deletion in *Lgr5* + ISCs. $Ccn1^{\Delta Lgr5}$ mice treated with TM showed markedly

stronger signals of non-phosphorylated, active β-catenin by immunohistochemistry and immunoblotting (Fig. 4d, e). Immunofluorescence staining of Axin2, a Wnt/β-catenin target gene[37], reinforced enhanced Wnt signaling in the crypts of $Ccn1^{\Delta Lgr5}$ mice with TM (Supplementary Fig. S4a). Accordingly, qPCR analysis showed that several Wnt target genes (*Axin2, Cd44, Ephb2, and Myc*) were upregulated (Fig. 4f) and

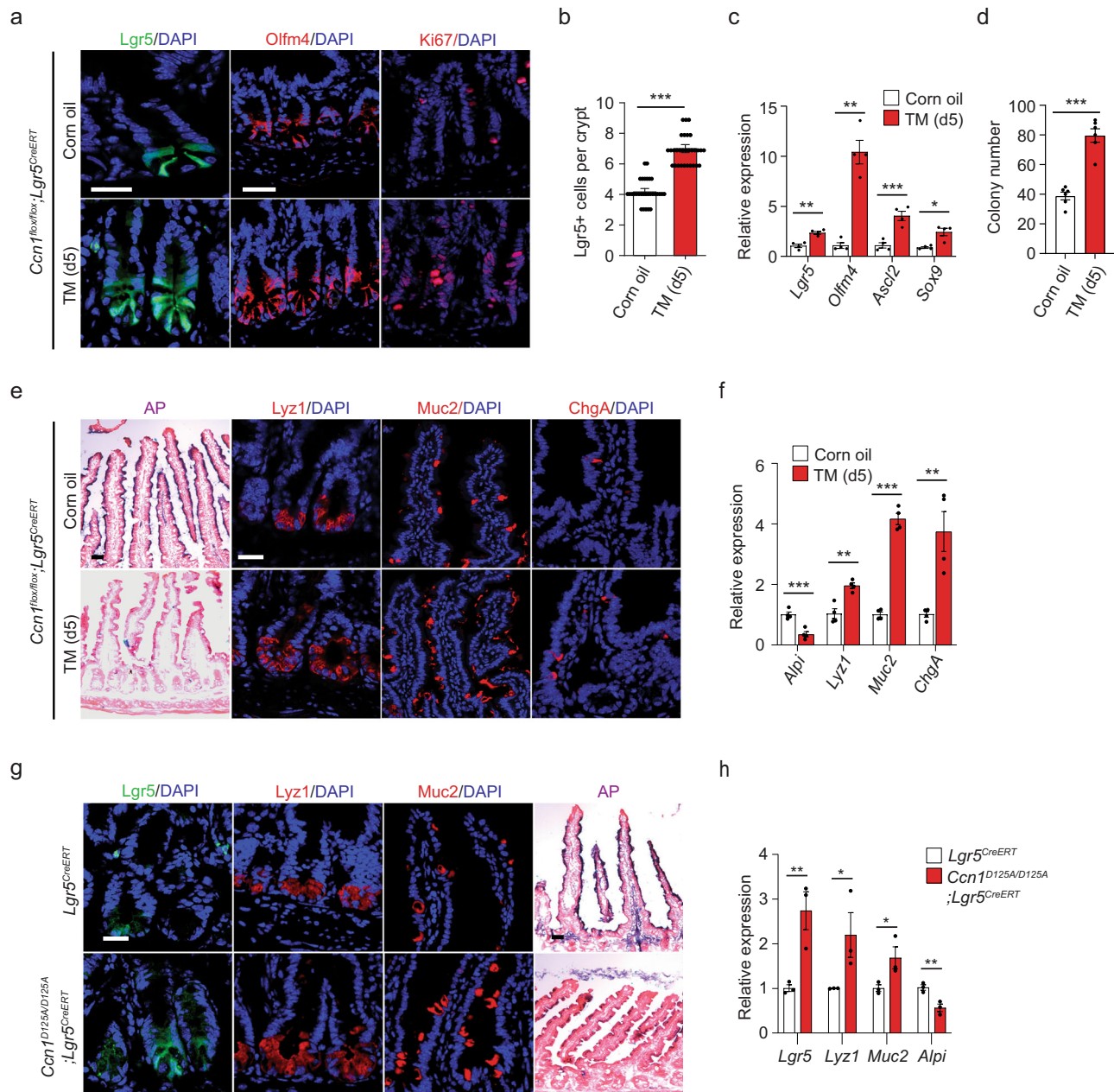

**Fig. 3 *Ccn1* deficiency leads to the ISC expansion, and increased secretory cells, but reduced enterocytes.** *Ccn1^ΔLgr5* mice were analyzed at day 5-post corn oil or TM. **a** Representative immunofluorescence images of *Lgr5* + ISCs (GFP), Olfm4, and Ki67 from at least three independent replicates. DAPI is counterstaining. Bars: 20 μm. **b** The average number of GFP + cells per crypt (~30 crypts from at least n = 4 mice per group). Data represent mean ± SEM. One-sided *t*-test. ***p < 0.0001. **c** qPCR analysis of ISC marker genes (*Lgr5, Olfm4, Ascl2,* and *Sox9*). Data represent the mean ± SEM of four biological replicates. One-sided *t*-test. *p < 0.05; **p < 0.01; ***p < 0.001. **d** in vitro colony formation assay of the crypt isolated from *Ccn1^ΔLgr5* mice with corn oil or TM (day 5). Data represent the mean ± SEM of six biological replicates. One-sided *t*-test. ***p < 0.001. **e** Alkaline phosphatase (AP) staining is to mark enterocytes. Immunofluorescence staining of Lyz1 (Paneth cells), Muc2 (goblet cells), and ChgA (enteroendocrine cells). Representative images are shown from three independent replicates. Bars: 20 μm. **f** qPCR analysis of differentiation marker genes (*Alpi, Lyz1, Muc2,* and *ChgA*). Data represent the mean ± SEM of four biological replicates. One-sided *t*-test. **p < 0.01; ***p < 0.001. **g** Immunofluorescence staining (*Lgr5* + GFP, Lyz1, and Muc2) and AP staining on the jejunum sections of *Lgr5^CreERT* and *Ccn1^D125A/D125A; Lgr5^CreERT* knock-in mice (n = 3 each). Images are representative of three replicates. Bars: 20 μm. **h** qPCR analysis of differentiation marker genes in the jejunum of *Lgr5^CreERT* and *Ccn1^D125A/D125A; Lgr5^CreERT* knock-in mice. Data represent the mean ± SEM of three biological replicates. One-sided *t*-test. *p < 0.05; **p < 0.01. Source data and the exact *p*-values are provided in a Source Data file.

the expression of Wnt ligand *Wnt3a*, but not the non-canonical ligands *Wnt5a* and *Wnt5b*, was increased (Fig. 4g). Interestingly, the expression of *Dkk1*, a secreted Wnt antagonist that inhibits canonical Wnt signaling by competitive binding to co-receptors LRP5/6 (ref. [38]), was greatly reduced (Fig. 4g), possibly contributing to enhanced Wnt signaling.

We further examined whether CCN1 regulation of both signalings in ISCs could be spatially identified in the crypts by double immunofluorescence staining of the activation marker and target genes of each signaling (Supplementary Fig. S4). The expression of Hes1 and Axin2 changed in both the level and pattern upon *Ccn1* deletion (Supplementary Fig. S4a), although

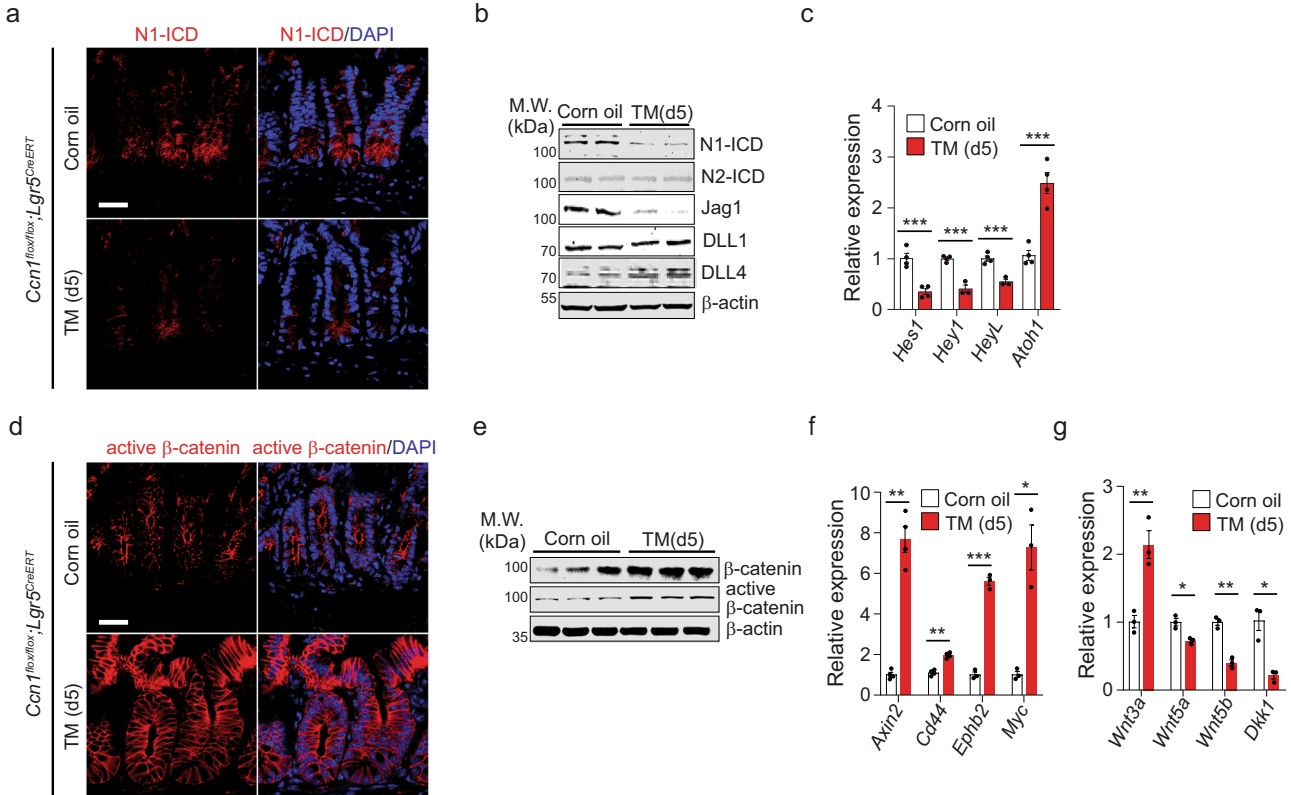

**Fig. 4 Reduced Notch signaling, but increased Wnt signaling in *Ccn1*-deficient mice.** *Ccn1^ΔLgr5* mice were analyzed at day 5-post corn oil or TM. **a** Immunofluorescence staining of Notch1 intracellular domain (N1-ICD, red) in the crypt of the jejunum. DAPI is counterstaining. Representative images are shown from three independent replicates. Bars: 20 μm. **b** Immunoblot analysis of indicated protein expression (N1-ICD, N2-ICD, Jag1, DLL1, and DLL2) in the jejunum. β-actin as a loading control. Representative images are shown from at least three replicates. **c** qPCR analysis of Notch target genes (*Hes1*, *Hey1*, and *HeyL*) and *Atoh1*. Data represent the mean ± SEM of four biological replicates. One-sided *t*-test. ***p < 0.001. **d** Immunofluorescence staining of active β-catenin in the crypt of *Ccn1^ΔLgr5* mice. Images are representatives of three independent replicates. Bars: 20 μm. **e** Immunoblot analysis of active and total β-catenin. Images are representative of three replicates. **f**, **g** qPCR analysis of Wnt target genes (*Axin2, Cd44, Ephb2,* and *Myc*; **f**), Wnt ligands (*Wnt3a, Wnt5a,* and *Wnt3b*; **g**), and a secreted Wnt antagonist *Dkk1* (**g**). Data represent the mean ± SEM of four biological replicates. One-sided *t*-test. *p < 0.05; **p < 0.01; ***p < 0.001. Source data and the exact *p*-values are provided in a Source Data file.

there was a regional separation in their expression. However, staining of N1-ICD with Sox9, another Wnt/β-catenin target gene in the intestine[39], identified a nice overlap in their expression in the crypt base (Supplementary Fig. S4b). Importantly, cells near the border of the stem cell compartment and TA zone exhibited high Sox9 expression but showed either reduction or lack of N1-ICD upon *Ccn1* deletion in *Lgr5* + ISCs, while most cells in the crypt bottom still expressed a modest level of N1-ICD and Sox9 (Supplementary Fig. S4b), indicating that there might be a positional difference in whether Notch and Wnt signaling would be engaged individually or simultaneously in ISCs of the crypts. Together, these results show that CCN1 regulates both Notch and Wnt signaling, thereby influencing fate decisions and proliferation of ISCs in the mouse intestine.

**CCN1-integrins α_vβ_3/α_vβ_5 signaling controlled ISC homeostasis.** Intestinal organoids retain the intestinal stem cell hierarchy and serve as an ideal system to study regulation and signaling mechanisms[40]. We cultured intestinal crypt organoids from *Ccn1^ΔLgr5* mice and deleted *Ccn1* in *Lgr5* + ISCs by treatment with 4-hydroxytamoxifen (4-OHT) (Supplementary Fig. S5a). These organoids recapitulated the compound phenotypes of *Ccn1^ΔLgr5* with TM, showing an increase in *Lgr5* + ISCs (GFP) and Ki67+ proliferating cells as well as Paneth cells (Lyz1) and goblet cells (Muc2), but a reduction in enterocytes as indicated by low AP activity (Fig. 5a, b).

Several lines of evidence indicated that CCN1 functions through integrins α_vβ_3/α_vβ_5. First, when we reintroduced recombinant CCN1 protein (from home-made or commercial source; see Methods) to these organoids, the phenotypes reverted to those of organoids without *Ccn1* deletion, but the addition of CCN1-D125A mutant protein unable to bind integrins α_vβ_3/α_vβ_5 had no effect (Fig. 5a, b, Supplementary Fig. S5b). Second, crypt organoids of *Ccn1^D125A/D125A* mice showed similar phenotypes: ISC expansion with increased Ki67+ cells, increased secretory cells, and reduced enterocytes (Supplementary Fig. S5c). Finally, a small-molecule inhibitor of integrins α_vβ_3/α_vβ_5 (SB273005) and an inhibitor of the integrin signaling mediator focal adhesion kinase (FAK; GSK2256098)[41], efficiently blocked the effects of CCN1 add-back in organoids (Fig. 5a, b). These results confirm that CCN1 regulates ISC proliferation and differentiation through integrins α_vβ_3/α_vβ_5.

**CCN1 promotes Notch signaling through NF-κB-mediated Jag1 expression.** *Ccn1^ΔLgr5* organoids after 4-OHT treatment suffered from reduced Notch signaling, as shown by diminished expression of Jag1, N1-ICD, and the Notch target genes *Hes1* (Fig. 5c, d). CCN1 add-back to these organoids restored Jag1 and N1-ICD levels as well as *Jag1* and *Hes1* expression (Fig. 5c, d). However, the expression of *Dll1* and *Dll4* was unaffected by *Ccn1* deletion or CCN1 add-back (Supplementary Fig. S6a). Moreover, the addition of a soluble Jag1 protein reversed the differentiation

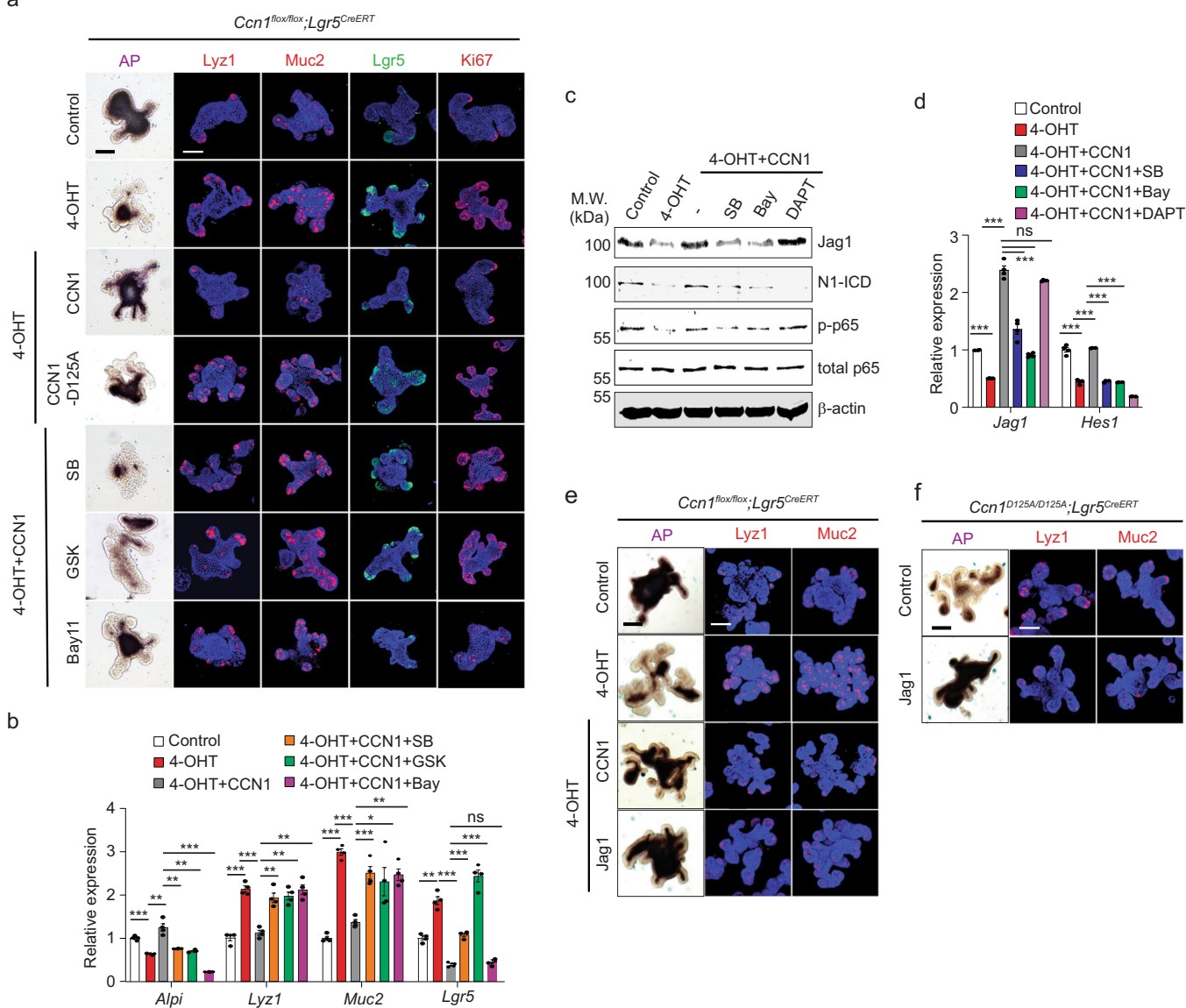

**Fig. 5 CCN1 regulates ISC differentiation through integrins $\alpha_v\beta_3/\alpha_v\beta_5$-mediated NF-κB activation. a** AP staining and immunofluorescence staining of markers (Lyz1, Muc2, $Lgr5 +$ GFP, Ki67) for the analysis of differentiation and proliferation of the $Ccn1^{\Delta Lgr5}$ organoids. DAPI is counterstaining. Organoids were treated with vehicle or 4-OHT (1 μM) for 5 days. For add-back experiments, CCN1 (WT) or CCN1-D125A proteins (4 μg per ml each) were treated to organoids 1 day after 4-OHT treatment began. Small-molecule inhibitors for integrin $\alpha_v$ (SB273005, 1 μM), FAK inhibitor (GSK2256098, 500 nM), and IκB kinase inhibitor (Bay11-7082, 5 μM) were pretreated 30 min before CCN1 add-back. Images are representative of three independent replicates. Bars: 100 μm. **b** qPCR analysis of lineage markers in organoids treated as in **a**. Data represent the mean ± SEM of four biological replicates. One-sided $t$-test. **$p < 0.01$; ***$p < 0.001$; ns not significant. **c, d** Organoids were treated as described in **a**. Inhibitors used are SB273005 (1 μM), Bay11-7082 (5 μM), and DAPT (10 μM). **c** Immunoblot analysis of p-p65, total p65, Jag1, and N1-ICD. Representative images are shown from three replicates. **d** qPCR analysis of $Jag1$ and $Hes1$ expression in organoids treated as in **c**. Data represent the mean ± SEM of four biological replicates. One-sided $t$-test. ***$p < 0.001$; ns not significant. **e, f** $Ccn1^{\Delta Lgr5}$ organoids with 4-OHT (1 μM, **e**) or $Ccn1^{D125A/D125A}$ organoids (**f**) were treated with CCN1 (4 μg per ml) or Jag1 (2 μg per ml) and examined for differentiation by AP staining and immunofluorescence staining of Lyz1 and Muc2. Bar: 100 μm. Representative images are shown from three independent replicates. Source data and the exact $p$-values are provided in a Source Data file.

phenotype of $Ccn1^{\Delta Lgr5}$ organoids treated with 4-OHT as well as organoids from $Ccn1^{D125A/D125A}$ knock-in mice (Fig. 5e, f). Thus, CCN1 regulates $Jag1$ expression in organoids and Jag1 is capable of mediating CCN1 effects in ISC differentiation.

Previously, CCN1 has been shown to induce NF-κB-dependent $Jag1$ expression to promote Notch signaling in cholangiocytes[22]. Likewise, the activation of NF-κ (p-p65/RelA) was reduced upon $Ccn1$ deletion in organoids but was restored by CCN1 add-back in immunoblot analysis (Fig. 5c). p-p65 levels were also significantly reduced in the crypts of TM-treated $Ccn1^{\Delta Lgr5}$ mice

(Supplementary Fig. S3b), indicating that CCN1 regulates NF-κB in the intestine. Importantly, a pharmacological IκB Kinase inhibitor (BAY11-7082) and the integrins $\alpha_v\beta_3/\alpha_v\beta_5$ inhibitor SB273005 efficiently blocked NF-κB activation (p-p65) and $Jag1$ induction by CCN1 add-back (Fig. 5c, d) and prevented the CCN1 add-back from reversing the effect of $Ccn1$ deletion on secretory cell differentiation (Fig. 5a). These inhibitors also reduced N1-ICD levels (Fig. 5c) and downregulated $Hes1$ expression (Fig. 5d), supporting the notion that CCN1 engagement of integrins $\alpha_v\beta_3/\alpha_v\beta_5$ triggers NF-κB activation, leading to

*Jag1* expression and Notch activation. By contrast, treatment with DAPT, a γ-secretase inhibitor that blocks Notch processing (N1-ICD release; Fig. 5c), inhibited *Hes1* expression despite NF-κB activation and *Jag1* induction upon CCN1 add-back (Fig. 5c, d). In addition, DAPT induced secretory cell differentiation in organoids but CCN1 could not reverse the DAPT effect (Supplementary Fig. S6b), confirming that Notch signaling is the downstream effector of CCN1. Together, these results strongly indicate that CCN1-integrins $\alpha_v\beta_3/\alpha_v\beta_5$ signaling leads to NF-κB-mediated induction of *Jag1*, which, in turn, activates Notch signaling and ensures the differentiation toward the absorptive enterocytes in intestinal organoids.

**CCN1 restricts ISC proliferation through the Src-YAP-Wnt axis.** CCN1 add-back reversed *Lgr5* + ISC expansion even in the presence of Bay11-7082 (Fig. 5a), suggesting that CCN1 regulation of ISC proliferation is independent of the NF-κB/Jag1/Notch pathway. Integrin-mediated signaling has been shown to activate FAK as a key signaling mediator, which recruits and forms a kinase complex with the non-receptor tyrosine kinase Src[41]. Importantly, the selective Src inhibitor dasatinib blocked the CCN1 add-back effect only on ISC proliferation (*Lgr5* + GFP and Ki67), but not differentiation (Fig. 6a, b), which contrasted with the FAK inhibitor (GSK2256098) that blocked both proliferation and differentiation effects of CCN1 add-back (Fig. 5a, b). These findings suggest that integrins $\alpha_v\beta_3/\alpha_v\beta_5$ signaling bifurcates into distinct pathways to regulate ISC proliferation and differentiation. Immunoblot analysis confirmed that Src (Y419) is activated in control organoids but not in organoids with *Ccn1* deletion (Fig. 6c). CCN1 add-back reactivated p-Src (Y419), which was blocked by SB273005 or dasatinib (Fig. 6c), demonstrating CCN1 activation of Src via integrin signaling to regulate ISC proliferation.

Among the signaling molecules that Src regulates in intestinal epithelial cells is YAP, a transcriptional co-activator important for tissue regeneration and tumorigenesis in the intestine[42,43]. Importantly, YAP is known to inhibit Wnt signaling and suppress ISC proliferation[44]. Src has been shown to activate YAP through direct phosphorylation of YAP (Y357)[45] or indirectly by phosphorylating, thus inactivating, the upstream negative regulator large tumor suppressor homolog 1 (Lats1)[46]. In parallel with the activation status of Src (Y419), both total YAP and activated p-YAP (Y357) were reduced upon *Ccn1* deletion but increased by CCN1 add-back (Fig. 6c), whereas the levels of active p-Lats1 (T1079) and inactive p-YAP (S127) phosphorylated by Lats1/2 showed the opposite trend. Moreover, SB273005 or dasatinib blocked CCN1 add-back-induced YAP activation (Fig. 6c), demonstrating that CCN1-integrins $\alpha_v\beta_3/\alpha_v\beta_5$ signaling leads to Src-mediated YAP activation. Consistently, immunohistochemistry showed nuclear-enriched YAP expression in the crypt base of control *Ccn1*$^{\Delta Lgr5}$ mice, but TM treatment greatly reduced YAP signals in the entire crypts of *Ccn1*$^{\Delta Lgr5}$ mice (Supplementary Fig. S7a). Immunoblot analysis confirmed that active p-YAP (Y357) decreased upon TM treatment, while inactive p-YAP (S127) increased (Supplementary Fig. S7b). YAP involvement was further examined using Super-TDU[47], an inhibitor of YAP interaction with TEA domain transcription factors (TEADs) derived from the Tondu domain of Vestigial-like protein 4 (VGLL4) (Supplementary Fig. S8a). In *Ccn1*$^{\Delta Lgr5}$ organoids with 4-OHT, Super-TDU blocked CCN1 add-back-induced restriction of ISC expansion without affecting differentiation (Fig. 6a, b), indicating the requirement of YAP-mediated transcription in CCN1 effects on ISC proliferation. Moreover, the activation of Wnt signaling, judged by β-catenin accumulation, was inversely correlated with Src and YAP

activation (Fig. 6c), suggesting a negative regulation of the Src-YAP axis on Wnt signaling.

Whereas the expression of Wnt ligands (*Wnt3a, Wnt3b, Wnt5a*, and *Wnt9b*) remained unchanged in *Ccn1*$^{\Delta Lgr5}$ organoids after 4-OHT treatment by qPCR analysis (Fig. 6d), the expression of *Dkk1* was significantly reduced among the secreted Wnt antagonists (*Dkk1, Dkk2*, and secreted frizzled-related protein 5; *sFrp5*; Fig. 6d), consistent with observations in vivo (Fig. 4g). As *Dkk1* is a YAP target gene[48], we hypothesized that CCN1 regulation of Wnt signaling might be through YAP-dependent *Dkk1* expression. Indeed, CCN1 add-back to 4-OHT-treated *Ccn1*$^{\Delta Lgr5}$ organoids rescued *Dkk1* expression, but Super-TDU efficiently blocked CCN1-induced *Dkk1* expression (Fig. 6e), resulting in the accumulation of β-catenin (Fig. 6f). SB273005 and dasatinib also blocked *Dkk1* induction upon CCN1 add-back (Supplementary Fig. S8b, c). Consequently, the expression of Wnt target genes (*Axin2, CD44, Ephb2*, and *Sox9*), repressed upon CCN1 add-back in 4-OHT-treated *Ccn1*$^{\Delta Lgr5}$ organoids, was elevated again by SB273005, dasatinib, and Super-TDU (Fig. 6g), confirming that CCN1 suppresses Wnt signaling through the integrins $\alpha_v\beta_3/\alpha_v\beta_5$-Src-YAP axis. Finally, a neutralizing anti-Dkk1 antibody efficiently inhibited CCN1 add-back-induced repression of ISCs proliferation in 4-OHT-treated *Ccn1*$^{\Delta Lgr5}$ organoids (Fig. 6h), reinforcing that Dkk1 is an effector of CCN1-mediated Wnt regulation. Together, these results identify a signaling pathway involving CCN1 engagement of integrins $\alpha_v\beta_3/\alpha_v\beta_5$, leading to Src-mediated activation of YAP, which down-regulates Wnt signaling in ISCs through Dkk1.

**A regulatory loop between CCN1 and YAP during ISC homeostasis.** While CCN1 activates YAP through integrin-mediated activation of Src (Fig. 6), *Ccn1* itself is well-recognized as a prominent YAP target gene[49–51]. We also detected YAP enrichment on the proximal TEAD binding region (−115b) of the *Ccn1* promoter[50] by chromatin immunoprecipitation (ChIP)-qPCR analysis on intestinal crypts, which was markedly reduced upon administration of verteporfin that disrupts a YAP/TEAD interaction (Supplementary Fig. S9a). Consistently, *Ccn1* expression decreased upon verteporfin by qPCR analysis (Supplementary Fig. S9b), suggesting a regulatory loop between CCN1 and YAP in ISC homeostasis. To test this possibility, we created *Yap*$^{\Delta Lgr5}$ mice (*Yap*$^{flox/flox}$; *Lgr5*$^{EGFP-CreERT}$). Deletion of *Yap* in *Lgr5* + ISCs by TM treatment virtually abolished YAP expression in the crypts, as shown by immunohistochemistry and qPCR analysis, concomitant with a similarly large decrease in *Ccn1* expression (Fig. 7a, b). Importantly, *Yap* deletion in *Lgr5* + ISCs led to significant expansion of *Lgr5* + ISCs and increases in Paneth cells (Lyz1) and goblet cells (Muc2) but decreased enterocytes (AP), phenocopying mice with *Ccn1* deletion in *Lgr5* + ISCs (Fig. 7a, b). Likewise, intraperitoneally injected verteporfin abolished CCN1 expression throughout the intestinal tissues of control *Ccn1*$^{\Delta Lgr5}$ mice and also led to phenotypes similar to *Ccn1* or *Yap* deletion in *Lgr5* + ISCs (Supplementary Fig. S9c–e). Thus, YAP may potentially regulate intestinal homeostasis, in part, by transcriptionally activating *Ccn1*.

To determine whether YAP may act through CCN1, we examined the effect of CCN1 add-back in *Yap*$^{\Delta Lgr5}$ organoids with 4-OHT. 4-OHT-treated *Yap*$^{\Delta Lgr5}$ organoids also exhibited increased *Lgr5* + GFP and Ki67 as well as secretory cell lineage markers Lyz1 and Muc2 by immunohistochemistry and qPCR analysis (Fig. 7c, d). Importantly, CCN1 add-back to these organoids only promoted the reversal of differentiation phenotype with a reduction in Lyz1 and Muc2 expression similar to those in control organoids, but *Lgr5* + GFP and Ki67 levels were

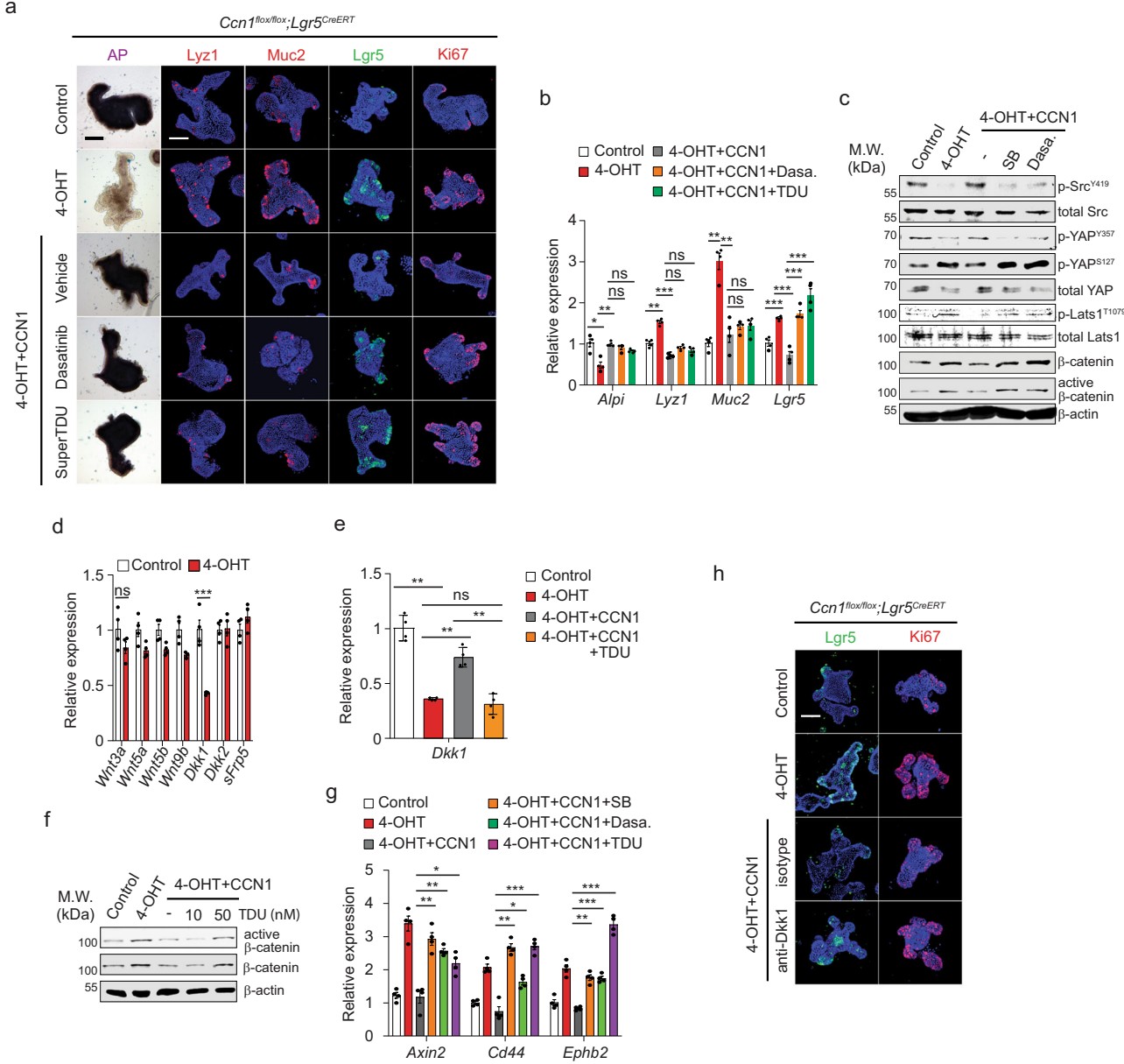

**Fig. 6 CCN1 regulates Wnt activity through integrins $\alpha_v\beta_3/\alpha_v\beta_5$-Src-YAP axis. a** AP staining and immunofluorescence staining of markers (Lyz1, Muc2, Lgr5 + GFP, and Ki67) for the analysis of differentiation and proliferation of the $Ccn1^{\Delta Lgr5}$ organoids. $Ccn1^{\Delta Lgr5}$ organoids were treated with vehicle or 4-OHT (1 μM) for 5 days. CCN1 (4 μg per ml) was added to organoids 1 day after 4-OHT. Inhibitors (Dasatinib, 10 nM; Super-TDU, 50 nM) were pretreated 30 min before CCN1. Representative images are shown from three independent replicates. Bars: 100 μm. **b** qPCR analysis of lineage markers in organoids treated as in **a**. Data represent the mean ± SEM of four biological replicates. One-sided *t*-test. *$p < 0.05$; **$p < 0.01$; *** $p < 0.001$; ns not significant. **c** Immunoblot analysis of indicated proteins from $Ccn1^{\Delta Lgr5}$ organoids treated as described in **a**. Representative images are shown from three replicates. **d** qPCR analysis of Wnt ligands (*Wnt3a, Wnt5a, Wnt5b,* and *Wnt9b*) and Wnt antagonists (*Dkk1, Dkk2,* and *sFRP5*) in $Ccn1^{\Delta Lgr5}$ organoids treated with vehicle or 4-OHT (1 μM, d5). Data represent the mean ± SEM of four biological replicates. One-sided *t*-test. ***$p < 0.001$; ns not significant. **e** qPCR analysis of *Dkk1* expression in $Ccn1^{\Delta Lgr5}$ organoids treated as indicated (CCN1, 4 μg per ml; Super-TDU, 50 nM). Data represent the mean ± SEM of four biological replicates. One-sided *t*-test. **$p < 0.01$; ns not significant. **f** Immunoblot analysis of active and total β-catenin in $Ccn1^{\Delta Lgr5}$ organoids treated as indicated (CCN1, 4 μg per ml; Super-TDU, 10 and 50 nM). β-actin as a loading control. Representative images are shown from three replicates. **g** qPCR analysis of Wnt target genes (*Axin2, Cd44,* and *Ephb2*) in $Ccn1^{\Delta Lgr5}$ organoids treated as indicated. Data represent the mean ± SEM of four biological replicates. One-sided *t*-test. *$p < 0.05$; **$p < 0.01$; ***$p < 0.001$. **h** Immunofluorescence staining of *Lgr5* + GFP and Ki67 in $Ccn1^{\Delta Lgr5}$ organoids treated as indicated. A neutralizing anti-Dkk1 antibody (5 μg per ml) or isotype control IgG was treated in place of inhibitors as in **a**. Representative images are shown from three replicates. Bars: 100 μm. Source data and the exact *p*-values are provided in a Source Data file.

still elevated (Fig. 7c). Corresponding changes in the expression of marker genes were shown by qPCR analysis, along with an increase in the Notch target *Hes1*, but not the YAP target *Dkk1*, upon CCN1 add-back (Fig. 7d). Consistently, immunoblot analysis detected that CCN1 add-back increased Jag1 and N1-

ICD, but failed to block β-catenin accumulation without YAP (Fig. 7e). These results imply that CCN1 still can activate Notch signaling for ISC differentiation independent of YAP, but YAP is indispensable for CCN1 to control ISC proliferation. By contrast, a soluble Dkk1 protein efficiently lowered the expression of

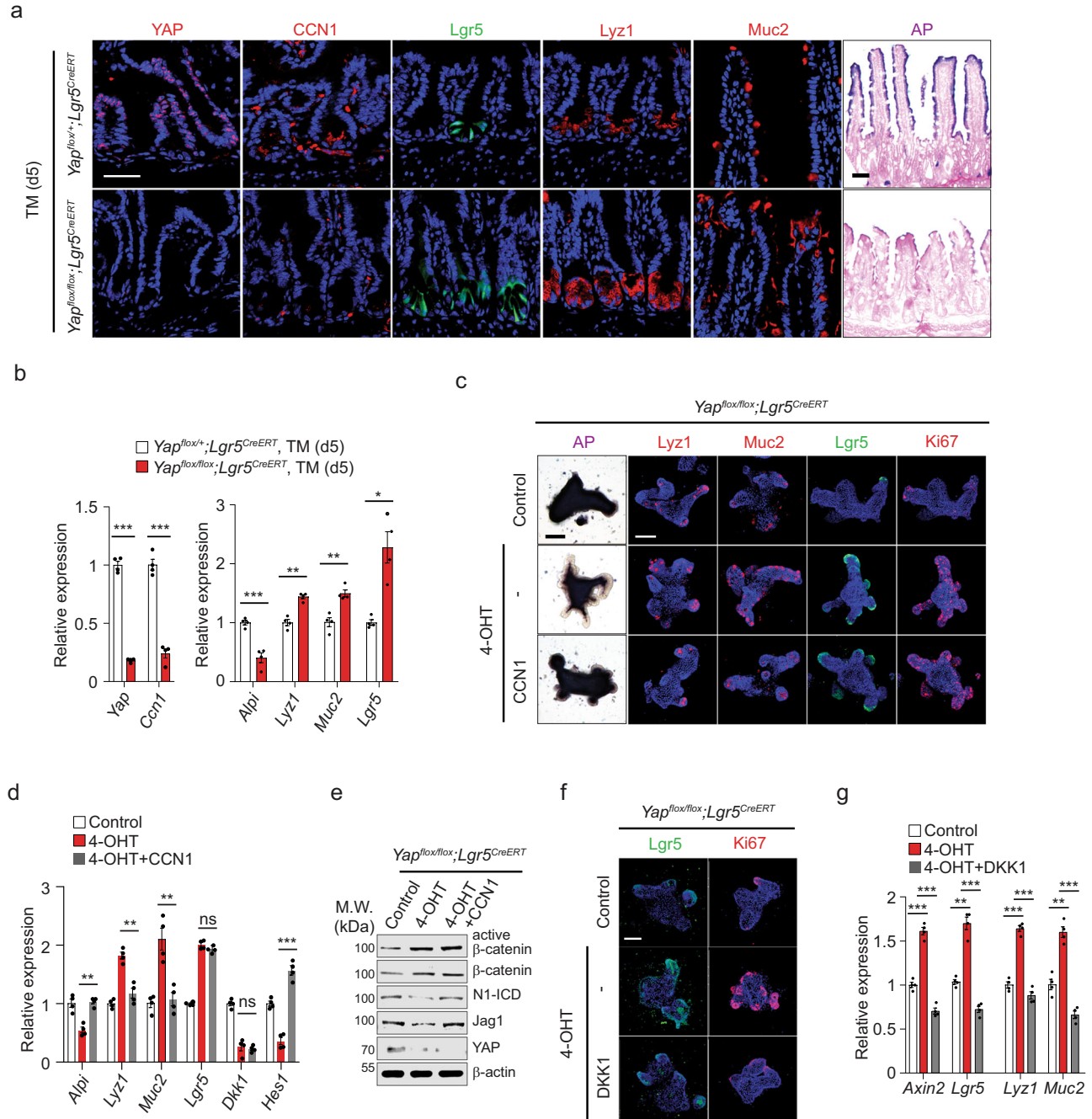

**Fig. 7 A regulatory loop between CCN1 and YAP during ISC homeostasis. a**, **b** $Yap^{flox/+}$; $Lgr5^{CreERT}$ or $Yap^{flox/flox}$; $Lgr5^{CreERT}$ ($Yap^{\Delta Lgr5}$) mice treated with corn oil or TM (day 5, $n = 4$ each). **a** Immunofluorescence staining of YAP, CCN1, and lineage markers ($Lgr5$ + GFP, Lyz1, and Muc2), and AP staining on the jejunum sections. Images are representative of three independent replicates. Bars: 20 µm. **b** qPCR analysis of indicated genes. Data represent the mean ± SEM of four biological replicates. One-sided $t$-test. *$p < 0.05$; **$p < 0.01$; ***$p < 0.001$. **c–e** $Yap^{\Delta Lgr5}$ organoids were treated with 4-OHT (1 µM), followed by CCN1 protein (4 µg per ml). **c** AP staining and immunofluorescence staining of Lyz1, Muc2, $Lgr5$ + GFP, and Ki67. DAPI is counterstaining. Images are representative of three replicates. Bars: 100 µm. **d** qPCR analysis of indicated genes. Data represent the mean ± SEM of four biological replicates. One-sided $t$-test. **$p < 0.01$; ***$p < 0.001$; ns not significant. **e** Immunoblot analysis of indicated proteins. Representative images are shown from three replicates. **f**, **g** $Yap^{\Delta Lgr5}$ organoids were treated with 4-OHT (1 µM), followed by Dkk1 proteins (50 ng per ml). **f** Immunofluorescence staining of $Lgr5$ + GFP and Ki67. DAPI is counterstaining. Representative images are shown from three independent replicates. Bar: 100 µm. **g** qPCR analysis on $Axin2$, $Lgr5$, $Lyz1$, and $Muc2$. Data represent the mean ± SEM of four biological replicates. One-sided $t$-test. **$p < 0.01$; ***$p < 0.001$. Source data and the exact $p$-values are provided in a Source Data file.

$Lgr5$ + GFP, Ki67, and a Wnt target $Axin2$ in 4-OHT-treated $Yap^{\Delta Lgr5}$ organoids (Fig. 7f, g), thus confirming that YAP acts through Dkk1 to inhibit Wnt signaling and $Lgr5$ + ISC proliferation. Interestingly, the elevated expression of $Lyz1$ and $Muc2$ in 4-OHT-treated $Yap^{\Delta Lgr5}$ organoids was also reduced by

the treatment of Dkk1(Fig. 7g), suggesting that Wnt inhibition may influence Notch signaling in the context of $Yap$ deletion. Together, these results indicate that the loss of $Ccn1$ expression upon $Yap$ deletion may be responsible for the increased secretory cell differentiation, and CCN1 regulation of ISC proliferation

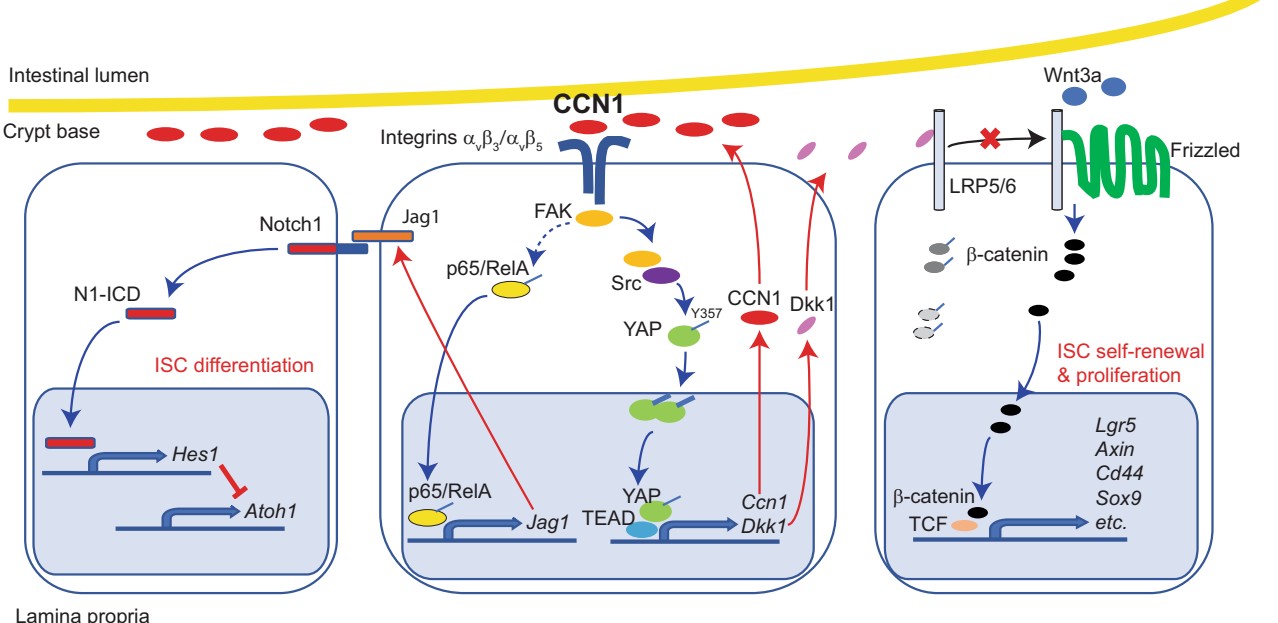

**Fig. 8 A diagram of CCN1 regulating Notch and Wnt signaling through distinct integrins $\alpha_v\beta_3/\alpha_v\beta_5$-mediated signaling pathways.** CCN1 expressed at the luminal side of the crypt base binds to integrins $\alpha_v\beta_3/\alpha_v\beta_5$ on, presumably, Paneth cells and *Lgr5* + ISCs, which elicits two distinct signaling pathways bifurcating downstream of FAK. For the regulation of Wnt signaling, FAK recruits and activates Src, leading to YAP stabilization and activation through phosphorylation at tyrosine 357. Active p-YAP (Y357) can enter the nucleus, where it promotes TEADs-dependent transcription of target genes, such as a secreted Wnt antagonist *Dkk1*, which interferes with Wnt binding to its cognate receptor Frizzled by sequestering a Wnt co-receptor LRP5/6. Concurrently, YAP-TEADs can further increase the expression of another target gene *Ccn1* in the crypt, suggesting a regulatory loop between CCN1 and YAP. On the other hand, integrin-FAK signaling results in NF-κB activation and subsequent induction of Jag1, a Notch ligand, which promotes Notch signaling important for ISC fate decision in the crypt base.

depends on YAP in a regulatory loop (Fig. 8 and Supplementary Fig. S11).

## Discussion

Homeostatic regeneration of the intestinal epithelium is achieved through the coordinated control of ISCs by their surrounding niche for self-renewal and differentiation into mature epithelial cells. However, various niche factors—soluble[52–55], membrane-bound[56], or extracellular matrix-associated[57]—were individually examined to induce signaling pathways solely designated for either proliferation or differentiation of ISCs and the factors or their mechanisms of action that exert coordinated regulation of ISCs proliferation and differentiation began to be appreciated recently. Here, we have identified the matricellular protein CCN1 as a niche factor that coordinately orchestrates both proliferation and fate decision of ISCs in the intestinal crypts by engagement of integrins $\alpha_v\beta_3/\alpha_v\beta_5$, which elicits distinct downstream pathways to regulate Notch and Wnt signaling (Fig. 8 and Supplementary Fig. S11). These findings provide insights into how a niche factor can deploy matrix signaling to exert overarching control of ISC homeostasis.

As a matricellular protein, CCN1 influences diverse functions of cells in the vicinity of the originating cells and the nearby extracellular matrix (ECM), as illustrated in the context of wound healing and tissue injury repair[58]. Here, we show that *Ccn1* is expressed from *Lgr5* + ISCs at the crypt base of the normal intestine and possibly affects both ISCs and Paneth cells for ISC homeostasis (Figs. 1, 3). Especially, CCN1 regulation of ISC homeostasis is through direct binding to two $\alpha_v$ integrins, $\alpha_v\beta_3$ and $\alpha_v\beta_5$, through a non-RGD sequence located in its von Willebrand factor type C (vWC) domain[59]. $Ccn1^{D125A/D125A}$ knock-in

mice expressing a mutant CCN1 unable to bind integrins $\alpha_v\beta_3/\alpha_v\beta_5$ phenocopy mice with *Ccn1* deletion in *Lgr5* + ISCs (Fig. 3g, h). These in vivo results, together with organoid experiments using the CCN1-D125A mutant protein and the $\alpha_v\beta_3/\alpha_v\beta_5$ antagonist SB273005 (Fig. 5), further establish that CCN1 acts through integrins $\alpha_v\beta_3/\alpha_v\beta_5$ to regulate proliferation and differentiation of ISCs. Several integrins are reported to be enriched at the intestinal crypt base, suggesting their potential role in ISC regulation[60]. Indeed, conditional deletion of $\beta_1$ integrins in intestinal epithelial cells caused crypt hyperplasia and dysplasia, although specific ECM molecules involved were not identified[61]. By contrast, our study uncovers both CCN1 and integrins $\alpha_v\beta_3/\alpha_v\beta_5$ as previously unknown regulators of ISCs and their specific activities as well as mechanisms of action in ISC homeostasis. Although *Lgr5* + stem cells exist in various tissues, exhibiting important roles in homeostasis and regeneration[62], whether CCN1 would have similar regulatory roles in stem cells of those tissues remains elusive.

CCN1 is involved in ISC fate decision by regulating Notch signaling, whose blockage results in skewed differentiation into secretory cells[16]. The jejunum of mice and intestinal organoids with *Ccn1* deletion in *Lgr5* + ISCs showed a reduction in N1-ICD release, its target gene *Hes1* expression, and subsequent derepression of *Atoh1* (Figs. 4a–c, 5), resulting in the increase in cells of secretory lineages at the expense of enterocytes (Figs. 3e, f, 5a, b). Moreover, CCN1 add-back to organoids with *Ccn1* deletion in ISCs, together with antagonists of integrins $\alpha_v\beta_3/\alpha_v\beta_5$, FAK, NF-κB, and γ-secretase, delineated signaling pathways and verified the importance of Notch signaling (Figs. 5, 8, and Supplementary Figs. S6, S11). Of particular interest is that Jag1, whose expression was controlled by CCN1, contributed to Notch activation and ISC fate decision, while DLL1 and DLL4 levels were mostly unaffected

(Figs. 4b, 5c–f, Supplementary Fig. S6a). Although previous gene deletion studies in mice have indicated that DLL1 and DLL4, rather than Jag1, play key roles in activating Notch signaling during ISC homeostasis[32,63], various Notch receptors and ligands appear functionally redundant in the intestine to safeguard the control of differentiation[32,63–65]. Thus, we speculate that CCN1-regulated *Jag1* expression might provide functional redundancy to ensure the potency of the absorptive cell fate, especially when Notch signaling might be weakened by altered expression of DLLs[32] or Notch glycosylation enzymes[34,35]. In our studies, the marked reduction of MFNG N-acetylglucosaminyltransferase in the intestinal crypt with *Ccn1* deletion in *Lgr5* + ISCs (Supplementary Fig. S3a) might render Notch1 activation dependent upon Jag1 (Figs. 4, 5). The regulation of MFNG expression is not well understood and how it might be controlled upon *Ccn1* deletion is currently unclear. Recently, excessive Wnt signaling has been linked to MFNG deficiency, imposing Jag1-dependent Notch activation[36]. Similarly, we observed that wild-type intestinal organoids exhibited reduced MFNG expression upon the repeated treatment of CHIR99021, a glycogen synthase kinase 3 (GSK3) inhibitor/Wnt activator[66] (Supplementary Fig. S10). Moreover, a neutralizing anti-Jag1 antibody in combination with CHIR99021, but not by itself, diminished Notch activation and led to increased secretory cell differentiation (Supplementary Fig. S10), reminiscent of phenotypes of *Ccn1* deletion. In addition, enhanced secretory cell differentiation in organoids with loss of *Ccn1* (*Yap* deletion; Fig. 6a), was blocked with Wnt signaling inhibition by Dkk1 treatment (Fig. 7g). Whether inhibition of Wnt signaling may restore MFNG expression and Notch signaling needs further examination.

CCN1 regulation of Wnt signaling, on the other hand, is through integrins $\alpha_v\beta_3/\alpha_v\beta_5$-FAK/Src-YAP axis (Fig. 6 and Supplementary Fig. S11), modulating ISC expansion during homeostasis (Fig. 3a, g). Mechanistically, CCN1 engagement and clustering of integrins $\alpha_v\beta_3/\alpha_v\beta_5$ lead to the recruitment of Src homology 2 (SH2) domain of Src and its conformational activation by phosphorylation at Y419 (ref. [41]). Active p-Src (Y419), in turn, phosphorylates YAP at Y357 (Fig. 6c), resulting in its stabilization and enhanced nuclear translocation (Supplementary Fig. S7)[67]. Src family kinase (SFK)-driven YAP activation has been shown in several cell types[45,68] and our data identifies CCN1 as a key YAP activator in ISCs. Importantly, YAP regulation of Wnt signaling is through the expression of Wnt antagonist Dkk1, a YAP target gene[48]. Thus, *Ccn1* deletion rendered decreased p-Src (Y319), reduction of p-YAP (Y357; Fig. 6c), and reduced *Dkk1* expression (Fig. 6d, e, and Supplementary Fig. S8b, c), resulting in enhanced Wnt signaling (Figs. 4d–f, 6c, and 6g) and exuberant expansion of *Lgr5* + ISCs in mice (Fig. 3a, g) and organoids (Figs. 5, 6). Several mechanisms have been proposed on how YAP regulates Wnt signaling. However, they all described the transcriptionally inactive, cytosolic YAP for this function by sequestering β-catenin in the cytosol[69] or binding and limiting a positive Wnt regulator Dishevelled (DVL)[44]. By contrast, our data show that transcriptionally active YAP in normal crypts and organoids regulated by CCN1-integrins $\alpha_v\beta_3/\alpha_v\beta_5$ signaling controls Wnt signaling and restricts ISC proliferation, most likely through the activation of the YAP target gene *Dkk1*.

Another important finding is that CCN1 and YAP positively regulate each other in a regulatory loop (Supplementary Fig. S11), and thus mice with deletion of *Ccn1* or *Yap* in *Lgr5* + ISCs essentially phenocopied each other in the regulation of ISCs (Figs. 3, 7). As *Ccn1* expression in the intestinal crypts was under transcriptional control of YAP during homeostasis (Supplementary Fig. S9a, b), CCN1 activation of YAP (Y357) through FAK-Src downstream of integrins $\alpha_v\beta_3/\alpha_v\beta_5$ (Fig. 6 and Supplementary Figs. S7, S8) is not only critical for Wnt regulation but also reinforces *Ccn1* expression. Conversely, YAP could maintain its active state and exert its regulatory functions in ISCs, in part, through the integrins $\alpha_v\beta_3/\alpha_v\beta_5$ signaling by promoting the expression of *Ccn1* as a prominent target gene. A similar CCN1-YAP regulatory loop was also found during tip cell fate decisions in the retina[51]. YAP likely deploys a regulatory loop with key regulatory molecules in the ECM, thus engaging matrix-driven signaling pathways for securing its expression and functions. Indeed, YAP has also been shown in a regulatory loop with CCN2 (CTGF), another CCN protein family member, in the mouse kidney and retina[70,71]. In addition to the homeostatic roles, the CCN1-YAP regulatory loop is expected ultimately to link CCN1 functions to intestinal regeneration after ISC injury. *Yap* expression changes dramatically during a dextran sulfate sodium (DSS) injury and regeneration in mice[43,72], which is reminiscent of *Ccn1* expression[24]. Based on the observation that mice with *Yap* deletion in the intestinal epithelium suffered 100% mortality with impaired epithelial regeneration following DSS injury[43], we may anticipate a similar phenotype in mice with *Ccn1* deletion in *Lgr5* + ISCs or knock-in mutation (*Ccn1*[D125A/D125A]) upon DSS injury and differentiate the roles of CCN1 that are mediated through distinct integrins ($\alpha_v\beta_3/\alpha_v\beta_5$ in epithelial cells vs. $\alpha_6\beta_1$ in fibroblasts) in the intestine.

In summary, our finding of CCN1 as a matricellular niche factor that coordinately regulates ISC proliferation and differentiation in homeostasis through integrin-mediated signaling expands our understanding of stem cell biology. Given the importance of ISCs in injury repair and epithelial regeneration after damage, further investigation on whether CCN1 plays a role in such contexts through the regulation of ISCs is warranted and may uncover new insight into the regenerative biology of the intestine.

## Methods

**CCN1 proteins and reagents**. Recombinant CCN1 and CCN1-D125A mutant proteins were produced using a baculovirus expression system in Sf9 insect cells and purified by ion exchange or immuno-affinity chromatography[73,74]. Sf9 cells (obtained from American Type Culture Collection, CRL-1711™) were routinely tested for mycoplasma contamination using LookOut® Mycoplasma PCR detection kit (Sigma). Purified CCN1 proteins were regularly examined for purity and contamination by SDS-gel electrophoresis and Limulus amoebocyte lysate method (for endotoxin) and also tested for specific activities in comparison with commercially available CCN1 protein (CCN1-Fc chimera protein; R&D systems, 4055-CR; Supplementary Fig. S5B). Integrin $\alpha_v\beta_3/\alpha_v\beta_5$ inhibitor (SB273005; S7540), NF-κB inhibitor (Bay11-7082; S2913), γ-secretase inhibitor (DAPT; S2215), YAP-TEAD association inhibitors (Super-TDU; S8554, Verterporfin; S1786), FAK inhibitors (GSK2256098; S8523), and GSK3β inhibitor (CHIR99021; CT99021) were purchased from Selleckchem. Src inhibitor (Dasatinib; SML2589) was from Sigma–Aldrich. Recombinant Jag1 protein (599-JG) and Dkk1 protein (5897-DK) were from R&D systems. A neutralizing anti-Jag1 antibody (HMJ1-29) was from Invitrogen (16-3391-85) and an anti-Dkk1 antibody was from R&D systems (AF1096). 4′,6-diamidino-2-phenylindole (DAPI) solution (#62248) was from Thermo Fisher scientific.

**Mice**. Animal protocols were approved by the Institutional Animal Care and Use Committee of The University of Illinois at Chicago (ACC#20-178). Transgenic CCN1(CYR61):EGFP (*Ccn1*[EGFP/+]) reporter mice in FVB/N-Swiss webster background were obtained from the Mutant Mouse Regional Resource Centers at the University of California, Davis[75]. Lgr5-EGFP-IRES-creERT2 knock-in mice were obtained from Jackson Laboratory (#08875)[2]. *Ccn1*[flox/flox] mice were constructed as described previously[76]. *Yap*[floxl/flox] mice were kindly provided by Dr. Duojia Pan[77]. *Ccn1*[ΔLgr5] and *Yap*[ΔLgr5] mice were generated by crossing *Ccn1*[flox/flox] or *Yap*[floxl/flox] mice with the Lgr5-EGFP-IRES-creERT2 mice. *Ccn1*[D125A/D125A] knock-in mice were constructed as described[20] and crossed with the Lgr5-EGFP-IRES-creERT2 mice for detecting *Lgr5* + ISCs in knock-in mice. All mice were housed in sterile static micro-isolator cages on autoclaved corncob bedding with water bottles. Both irradiated food and autoclaved water were provided ad libitum. The standard photoperiod was 14 h of light and 10 h of darkness. The housing facility maintained the temperature at 75 °F and the relative humidity of 40–60%. In our studies, both male and female mice at 2–4 months of age with similar bodyweight (25–28 g) were used. For *Cre* induction, mice were daily injected (i.p.) with 100 µl of tamoxifen (Sigma–Aldrich, T5648) in corn oil at 100 mg per ml.

**In vivo paracellular permeability assay.** $Ccn1^{\Delta Lgr5}$ mice at the post-injection (either corn oil or tamoxifen) day 10 were orally gavaged with fluorescein isothiocyanate (FITC)-dextran (MW 4000, 60 mg per kg; Sigma–Aldrich, #46944) 4 h before sacrifice. Blood was collected via intraorbital vein blood collection, and the fluorescence intensity in the serum was measured (excitation at 485 nm, emission at 520 nm) using a Victor3V plate reader with Victor3 WorkOut™ software version 1.5 (PerkinElmer). FITC-dextran concentrations were determined from a standard curve generated by a serial dilution.

**Measurement of glucose concentrations.** $Ccn1^{\Delta Lgr5}$ mice at the post-injection (either corn oil or tamoxifen) day 10 were fasted overnight and treated either with saline (vehicle) or sucrose (3 g per kg) by oral gavage. After the indicated time, Blood samples collected from the tail vein were assayed with a commercial glucose meter (Contour, Bayer).

**Intestinal organoid culture.** Mouse intestinal organoids were established and maintained as described previously[78]. Briefly, the freshly harvested small intestine (jejunum) was flushed with cold phosphate-buffered saline (PBS), cut longitudinally, and washed with cold PBS (3×). The washed intestine was further cut into small pieces (~5 mm) and incubated in ethylenediaminetetraacetic acid (EDTA, 2 mM) in PBS with rocking for 30 min at 4 °C. After vigorous suspension in cold PBS for 15 sec, the mixture was passed through a 70 μm cell strainer (BD Biosciences, #352350) and the crypt fraction was enriched through centrifugation (5 min at $350 \times g$). The crypts were then embedded in Matrigel (Corning, #356255), seeded on a 24-well plate, and allowed for polymerization at 37 °C for 15 min. IntestiCult™ Organoid Growth Medium (STEMCELL Technologies, #06005) was added and refreshed every 2 days. For passaging, the organoids embedded in Matrigel in each well were directly suspended in 1 ml of cold PBS after removal of medium and were pelleted by centrifugation (3 min at $350 \times g$). The pelleted organoids were embedded in fresh Matrigel and seeded on the plate, followed by the addition of fresh culture medium every 2 days. For in vitro colony formation assay, crypts were isolated from $Ccn1^{\Delta Lgr5}$ mice injected with either corn oil or tamoxifen on day 5. Approximately 100 crypts per well were seeded in a 24-well plate and organoid formation was counted on day 3. For induction of Cre activity, the organoids from $Ccn1^{\Delta Lgr5}$ or $Yap^{\Delta Lgr5}$ mice were cultured in presence of 4-hydroxytamoxifen (4-OHT, 1 μM; MiliporeSigma, H6278). Each inhibitor was added after 12 hr of 4-OHT treatment. Recombinant CCN1 or CCN1-D125A (4 μg per ml, each), Jag1 (2 μg per ml), or Dkk1 (2 μg per ml) was added to the culture medium on day 2. The organoids were harvested on day 5 for subsequent RNA or protein extraction. The concentrations of inhibitors and neutralizing antibodies used in organoids are as follows: integrin $\alpha_v$ inhibitor (SB273005, 1 μM), NF-κB inhibitor (Bay11-7082, 5 μM), γ-secretase inhibitor (DAPT, 10 μM), FAK inhibitor (GSK2256098, 500 nM), Src inhibitor (dasatinib, 10 nM), YAP-TEAD association inhibitor (Super-TDU, 50 μM), GSK3β inhibitor (CHIR99021, 5 μM), anti-Jag1 (3 μg per ml), and anti-Dkk1 (3 μg per ml).

**Immunoblot analysis.** Samples were prepared in dithiothreitol (DTT)-containing 4× Laemmli sample buffer (Bio-RAD, #1610747) and resolved on SDS-PAGE, which was transferred to a polyvinylidene difluoride (PVDF; Bio-RAD, #1620177) membrane, and immunoblot analysis was performed using standard protocols. Antibody lists are provided in Supplementary Table 1. After drying the membrane, fluorescence images were obtained on the Odyssey® CLx Imaging System with Image Studio™ Software V. 5.0 (LI-COR). Uncropped blots are provided in Supplementary information.

**RNA isolation and quantitative RT-PCR.** Total RNA was purified from jejunum tissue or intestinal organoids using the GeneJET RNA purification Kit (Thermo Scientific, K0732) following the manufacturer's protocol. Total RNAs (2 μg) were reverse transcribed using Superscript Reverse Transcriptase III (Invitrogen, #18080044). Quantitative RT-PCR was performed with iCycler Thermal Cycler with CFX manager software V2.1 (Bio-Rad) using iQ SYBR Green Supermix (Bio-Rad, #1708880). PCR specificity was confirmed by agarose gel electrophoresis and melting curve analysis. A housekeeping gene (cyclophilin E) was used as an internal standard. Gene-specific primers used are listed in Supplementary Table 2.

**Chromatin immunoprecipitation (ChIP).** ChIP analysis was performed on freshly isolated intestinal crypts pooled from three jejunal tissues of control $Ccn1^{\Delta Lgr5}$ mice treated with either verteporfin (30 mg per kg) or dimethyl sulfoxide (DMSO, vehicle) for 5 days. The crypts were isolated as above and DNA-protein cross-link was achieved by formaldehyde (1%) for 10 min at room temperature (RT). After quenching with glycine (125 mM) for 5 min, cells were rinsed with cold PBS (3×) and resuspended in buffer containing 50 mM HEPES (pH 7.5), 140 mM NaCl, 1 mM EDTA (pH 8.0), Triton X-100 (1%), Na-deoxycholate (0.1%), SDS (0.1%), and protease inhibitors (Roche). The chromatin was sheared using an Ultrasonic processor (W-385, Heat systems-Ultrasonics, Inc.) for 20 min with the following settings: high power, 30 s on, 30 s off, 20 cycles. The average fragments of 200–500 bp were confirmed on a 1.5% agarose gel. For each ChIP, equal amounts of sample (25 μg) were incubated with an anti-YAP antibody (Cell Signaling Technology, # 4912; 2 μg) or isotype control IgG (rabbit) for 12 h at 4 °C, followed

by protein A/G agarose beads (Millipore, 20 μl) for additional 3 h. The beads were rinsed three times (3×) with wash buffer 1 containing 20 mM Tris–HCl (pH 8.0), 0.15 M NaCl, 2 mM EDTA (pH 8.0), 0.1% SDS, and 1% Triton X-100 and once with wash buffer 2 (the same buffer with 0.5 M NaCl). Chromatin was eluted by incubation of the beads with elution buffer (10 mM Tris–HCl pH 8.0, 0.3 M NaCl, 5 mM EDTA pH 8.0, 1% SDS, 1 μl RNaseA) at 65 °C overnight, and then purified using a QIAquick PCR purification kit (Qiagen). qPCR analysis was performed as above on 1 ng genomic DNA using primers designed for either the proximal (−115b) or distal (−2.1 kb) YAP-TEAD binding region of the $Ccn1$ promoter[50] and data were presented as a percentage of input.

**AP staining and immunohistochemistry.** Paraffin-embedded intestinal sections (5 μm) were stained with Hematoxylin/Eosin (H&E). Images were acquired using a Leica DM4000B microscope mounted with a QI Click CCD digital camera (QImaging) and analyzed for measuring the length of villus and crypt using ImageJ software (bundled with 64-bit Java 1.8.0_172, NIH). Alkaline phosphatase staining was performed using the NBT/BCIP Kit (Vector laboratories, SK-5400). Frozen intestinal sections (7 μm), fixed with ice-cold acetone, were probed with individual primary antibodies (listed in Supplementary Table 1) in immunofluorescence staining. DAPI (1 mg per ml) was used as a counterstain. Double immunofluorescence staining was performed by either serial incubation of the primary antibody from the same host with additional blocking in between or simultaneous incubation of two primary antibodies from different hosts and visualized with Alexa Fluor 546 goat anti-rabbit IgG/APC goat anti-rabbit IgG (serial) or Alexa Fluor 546 goat anti-rabbit IgG/Alexa Fluor 680 goat anti-mouse IgG (simultaneous). Fluorescence images were acquired using an LSM 700 confocal microscope with ZEN Black software version 8.1.0.484 (Zeiss) or Leica DM4000B microscope with Image-Pro Insight software V.8.0 (Media Cybernetics) and processed with Photoshop 2022 (Adobe). Fluorescent signals from APC goat anti-rabbit IgG or Alexa Fluor 680 goat anti-mouse IgG were converted to a pseudo-coloring of yellow.

For staining of whole-mount organoids, intestinal organoids were seeded in an eight-well glass-bottom chamber slide (Ibidi, #80827). The organoids were fixed in 4% paraformaldehyde (PFA) for 10 min and washed in PBS (3×) at RT. Following permeabilization with PBS containing 0.3% Triton X-100 (PBST) for 20 min RT and blocking in PBS containing 5% BSA (Sigma) for 1 h at RT, organoids were incubated with primary antibodies (listed in Supplementary Table 1) in PBS containing 2% BSA for overnight at 4 °C, followed by immunofluorescence staining, as described above.

**In situ hybridization.** Oligonucleotide probe for $Ccn1$ (5'- CTGCGGCTGCTGT AAGGTCTGCGCTAAACAACTCAA -3') was designed using OligoMinerApp (http://oligominerapp.org)[79] and synthesized with 3' Alexa Fluor 546 conjugation from Integrated DNA Technologies (IDT, Iowa). The probe was prepared at 100 nM working concentration in saline sodium citrate (SSC, pH 4.5) hybridization buffer containing 10% formamide and BSA (1 mg per ml), denatured at 65 °C for 10 min, and kept on ice before hybridization. Frozen sections (7 μm thick) were dried for 30 min, fixed with a 10-fold dilution of commercial formalin (37%) in PBS for 20 min, and permeabilized in PBST for 30 min at RT. After overnight hybridization at 30 °C, sections were serially washed with 4×, 2×, 1×, and 0.1× SSC (each containing 10% formamide) and counterstained with DAPI. Images were acquired using a confocal microscope (Zeiss, LSM700).

**Statistics and reproducibility.** All experiments were carried out with at least three biological replicates and showed successful reproducibility. All graphs were generated using GraphPad Prism9 software V.9.3.1. (Dotmatics) or Origin Pro 2020 (64-bit) SR1, V.9.7.0.188 (OrginLab Corp.) and data represent the mean ± SEM (standard error of the mean) with individual quantitative values in dots (column graph). Whenever necessary, statistical evaluation was performed by one-sided, two-sample with equal variance $t$-tests using the same software as above. $*p < 0.05$, $**p < 0.01$, and $***p < 0.001$ were considered significant. Exact $p$-values are provided in Source data. For immunoblotting, immunohistochemistry, and in situ hybridization, representative images are shown. Each of these experiments was independently repeated at least three times.

**Reporting summary.** Further information on research design is available in the Nature Research Reporting Summary linked to this article.

## Data availability

Data supporting the findings of this work are available within the main text and the Supplementary Information files. Source data used for tabulated representation are provided with this paper.

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

## Acknowledgements

We thank Dr. Duojia Pan (UT Southwestern) for the gift of *Yap^flox/flox* mice. This work was supported by a grant from the National Institutes of Health (DK108994; awarded to Dr. Lester F. Lau, University of Illinois at Chicago).

## Author contributions

J.H.W., J.S.C., and J.-I.J. designed and conducted the experiments; J.-I.J. analyzed the data and wrote the paper.

## Competing interests

The authors declare no competing interests.
