## [Peer Review File · Nature Communications]

CCN1 interacts with integrins to regulate intestinal stem cell proliferation and differentiationREVIEWER COMMENTS

Reviewer #1 (Remarks to the Author):

The results contained in the paper extend previous work from this group, albeit in a somewhat different system.

The data are largely of high quality and add to the idea that ccn proteins have context-dependent effects on modulating stem cell behavior via YAP.

1. The authors somewhat gloss over the fact that it is now, I think, generally appreciated that CCN proteins act in a proadhesive autocrine signaling loop via YAP. It is perhaps useful to formally show and emphasize that in this system YAP actually regulates CCN1 expression.
2. Recently published papers showing CCNs act in a YAP loop could also be discussed.
3. Several of the experiments rely on home-made CCN1. How do the authors know that the preparation is pure and, for example, free of TGFbeta contamination? Can similar results be obtained using commercially supplied rCCN1?

Reviewer #2 (Remarks to the Author):

In this paper, the molecular mechanism of CCN1 on the self-renewal and differentiation of ISCs in the crypt base in the intestine was studied. In particular the phenotype of the intestinal crypt was changed by deletion of CCN1 specifically in Lgr5+ cells, and the mechanism was investigated after finding decreased nutrient intake. And the involvement of integrin $\alpha\beta3/\alpha\beta5$ was confirmed using a mutant that CCN1 cannot bind to integrin, and Wnt signaling and Notch signaling already known to be involved in crypt homeostasis were examined. The association between each signaling and ISC proliferation or differentiation was studied in detail by various signaling inhibitors. Finally, it was concluded that Yap signaling maintains ISC homeostasis by feedback activation of CCN1 activity. In order to draw such a picture of this conclusion, a clear experimental methods were used as a well-planned strategy, and the meaning of each result was well interpreted.

However, if the authors can answer the questions below, it will be more improved and general interest to the readers.

1. Lgr5 is expressed in the other tissues, such as muscle cells. Did the same phenomena occur in other tissues Lgr5+CCN1 deleted cells as the ISC cells in the Lgr5+cell specific CCN1 deleted mouse? Is this an ISC specific mechanisms?
2. Among the downstream signaling activated by CCN1, if Jag-1-mediated Notch activation regulates ISC differentiation and src-mediated dkk1 activation regulates ISC proliferation, can this bipolarity occur between neighboring ISCs or one ISC in intestinal crypt? And what is the proportion of that population? Can this be verified spatially in the crypt?
3. Minor comments are 1) In Fig. 1G, the labeling of the x-axis and the point of the graph do not match. 2) the results of Fig 4E and F are missing in the text.

Reviewer #3 (Remarks to the Author):

Comments to Authors

The authors identified that CCN1, a matricellular protein, restricts Lgr5+ ISCs for intestinal homeostasis via controlling Notch and Wnt signaling, which unveiled the crucial roles of matrix signaling in intestinal homeostasis and possibly intestinal regeneration. Overall, the experiments were well designed and performed. The results support the authors' statement. The statistical analyses are appropriate.

Major comments

- The authors' previous study on CCN1 in intestinal regeneration (colitis) via IL6 (Cho et al., 2015) dampens the novelty of this study.
- It is recommended to include quantitative analysis of Ccn1-EGFP positive cells, villus length, and crypt depth of duodenum, jejunum, and ileum, and to briefly discuss why Ccn1 cKO-induced phenotype was only observed in the jejunum - maybe, due to the relatively higher expression of Ccn1 in the jejunum? Similarly, the expression pattern of integrins in the small intestine needs to be included.

Minor comments

- (Discussion) "We may anticipate a similar phenotype in mice with Ccn1 deletion or mutation (D125A) upon DSS injury". The authors' previous study (2015) already showed the IBD-related phenotype in Ccn1 ablated mice. Clarification and revision will help.
- It is recommended to tone down for 'knowledge gap' statement. Quite a few studies reported the importance of niche and non-cell-autonomous factors orchestrating intestinal homeostasis as well as regeneration - mainly using murine small intestine and organoids as model systems.

REVIEWER COMMENTS

→ We appreciate the reviewers' constructive comments and insightful questions. We responded to each comment below.

Reviewer #1 (Remarks to the Author):

The results contained in the paper extend previous work from this group, albeit in a somewhat different system.

The data are largely of high quality and add to the idea that ccn proteins have context-dependent effects on modulating stem cell behavior via YAP.

→ We thank the reviewer for the encouraging comments and also are glad that the reviewer appreciates the quality of the data.

1. The authors somewhat gloss over the fact that it is now, I think, generally appreciated that CCN proteins act in a proadhesive autocrine signaling loop via YAP. It is perhaps useful to formally show and emphasize that in this system YAP actually regulates CCN1 expression.

→ As the reviewer pointed out and we showed in the manuscript, YAP regulates CCN1 expression (Fig. 6) and CCN1, in turn, activates YAP through an integrin-mediated pathway (Fig. 5). We stated that CCN1 and YAP amplify the activity of each other in a regulatory loop in the abstract, result, and discussion. As suggested, we emphasized YAP regulation of CCN1 expression specifically in the intestine, which can be found in the discussion (p20).

2. Recently published papers showing CCNs act in a YAP loop could also be discussed.

→ We are also aware that there are a couple of recently published papers regarding the regulatory loop between CCN1 and YAP. We added these to the discussion with references (p20, Ref. 65 and 66).

3. Several of the experiments rely on home-made CCN1. How do the authors know that the preparation is pure and, for example, free of TGFbeta contamination? Can similar results be obtained using commercially supplied rCCN1?

→ We understand the concern raised by the reviewer, as the question of the potential contaminants is often a concern in describing novel activities of a protein. Given that protein preps purified from cells are seldom truly 100% pure, it is critical to determine whether there are contaminants that confound the results. In our studies, we are confident that the activities described in the manuscript are bona fide functions of CCN1 based on evidence from biochemical analysis and experimental results. 1) We routinely evaluate our protein preps using SDS-gel electrophoresis, which allowed us to estimate CCN1 proteins to be >90-95% pure. However, we do not exclude the possible minor impurities that are not visible on a gel. We also regularly test sf9 insect cells, viral stocks, and all related solutions. 2) In our organoid experiments (Fig. 4A), the add-back of CCN1-WT proteins, but not CCN1-D125A mutant proteins, reverted the phenotypes in *Ccn1*-deleted organoids, although they both are purified in the same way, suggesting that the effect would not be from the potential contamination. 3) Compared to

our sf9 insect cell production system, commercially supplied CCN1 proteins are from either bacterial expression systems or Fc-fusion proteins from mammalian cells. When testing them for comparison, we found that all CCN1 proteins showed no difference (slightly better response with our CCN1 preps) in the several activities tested so far (Jun and Lau, 2020, Nature Communications). For current studies, we tested again our CCN1 proteins and Fc-fusion CCN1 protein (from R&D systems) in *Ccn1*-deleted organoids and found they exhibit the same activity (restriction of *Lgr5*+ISC expansion). We provide the results below only for the reviewers. We also revised the Materials and Methods section to include these details regarding CCN1 protein preparation, quality controls, and their activities.

Only for reviewers' view

Activity comparison between CCN1 proteins from the house (4 μ g per ml) and FC-fusion CCN1 protein from R&D systems (4 μ g per ml) in reverting the *Lgr5*+ ISC expansion to control levels

Reviewer #2 (Remarks to the Author):

In this paper, the molecular mechanism of CCN1 on the self-renewal and differentiation of ISCs in the crypt base in the intestine was studied. In particular the phenotype of the intestinal crypt was changed by deletion of CCN1 specifically in *Lgr5*+ cells, and the mechanism was investigated after finding decreased nutrient intake. And the involvement of integrin $\alpha\beta3/\alpha\beta5$ was confirmed using a mutant that CCN1 cannot bind to integrin, and Wnt signaling and Notch signaling already known to be involved in crypt homeostasis were examined. The association between each signaling and ISC proliferation or differentiation was studied in detail by various signaling inhibitors. Finally, it was concluded that Yap signaling maintains ISC homeostasis by feedback activation of CCN1 activity. In order to draw such a picture of this conclusion, a clear experimental methods were used as a well-planned strategy, and the meaning of each result was well interpreted.

→ We are glad that the reviewer appreciated that our strategy was well-planned and the results were well-interpreted.

However, if the authors can answer the questions below, it will be more improved and general interest to the readers.

1. *Lgr5* is expressed in the other tissues, such as muscle cells. Did the same phenomena occur in other tissues *Lgr5*+CCN1 deleted cells as the ISC cells in the *Lgr5*+cell specific CCN1 deleted mouse? Is this an ISC specific mechanisms?

→ As the reviewer pointed out, *Lgr5* is broadly expressed in stem cells of many tissues, among which we focused on the intestinal epithelium that continuously renews and regenerates, thus requiring constant *Lgr5*⁺ stem cell activity during homeostasis. We have not examined other tissues as to whether the same CCN1 regulation exists on stem cells during homeostasis. However, stem cell activity in many tissues, including muscles, would be more critical in the context of damage/injury for repair and regeneration as shown in recent studies by Leung, C., *et al.* (2020), titled “*Lgr5* Marks Adult Progenitor Cells Contributing to Skeletal Muscle Regeneration and Sarcoma Formation”. Similarly, when we examined the tibialis anterior muscle of control mice and mice with *Ccn1* deletion in *Lgr5*⁺ stem cells, we could not find *Lgr5*⁺ signals (GFP) in muscle from either mouse (without injury). We speculate that the effect of *Ccn1* deletion in *Lgr5*⁺ stem cells in such tissues as muscle might emerge upon injury, but pursuing other tissues is beyond the scope of this manuscript. We included the discussion regarding *Lgr5*⁺ stem cells in other tissues (p18).

2. Among the downstream signaling activated by CCN1, if Jag-1-mediated Notch activation regulates ISC differentiation and src-mediated *dkk1* activation regulates ISC proliferation, can this bipolarity occur between neighboring ISCs or one ISC in intestinal crypt? And what is the proportion of that population? Can this be verified spatially in the crypt?

→ We appreciate the insightful questions from the reviewer. We also recognized that it is an important question whether CCN1 regulation of these two important signaling pathways could occur on the same ISC or between neighboring ISCs independently. We anticipated that the examination of activation markers (NICD for Notch activation) or target genes of each signaling (*Axin2* and *Sox9* for Wnt signaling and *Hes1* for Notch signaling) using double fluorescence staining would address this issue spatially in the crypt, as the localization of Wnt and Notch signaling in the intestinal crypts has been nicely demonstrated in studies by Tian *et al.* (2015; this reference is newly cited in the manuscript, Ref. 8). For this purpose, we have screened several antibodies for *Hes1* and *Axin2*, as there were no well-verified antibodies for histochemistry and, thus, many studies rather used reporter strains to monitor these proteins.

The followings are the results of the antibodies that we verified. 1) An anti-*Hes1* monoclonal antibody (Santacruz, sc-166378), among several tested antibodies, detects *Hes1* in the crypt epithelium, which shows, however, only a cytosolic pattern in the middle area of the crypts, but not in the crypt bottom. 2) A new anti-NICD polyclonal antibody (abcam, ab52301; replacing discontinued ab8925 used in Fig. 4A) stained NICD strongly in the crypt bottom and only weakly in the middle area. Upon *Ccn1* deletion, NICD signals in the middle area of the crypt became lost. 3) An anti-*Axin2* polyclonal antibody (abcam, ab107613) detected *Axin2* in the stem cell compartment of the crypt bottom with a punctuated pattern around the nucleus as well as the cytosol, and its expression was strongly extended toward cells near the border of the stem cell compartment and the TA zone upon *Ccn1* deletion. 4) A similar expression pattern was found in another Wnt target gene *Sox9* (EMD Millipore, AB5535), which exhibited the clearest staining enough to distinguish individual cells that express *Sox9*.

We have initially tested a combination of *Hes1* and *Axin2* in double immunofluorescence staining, but the spatial distinction and individual cell monitoring were difficult due to the expression patterns of these proteins (shown in new Supplementary Fig. S4A). Therefore, we decided to use NICD and *Sox9*, which showed that upon *Ccn1* deletion, cells near the border of the stem cell compartment particularly

expressed increased Sox9 (active Wnt) with reduced NICD in the same cell, suggesting that both the Notch and Wnt signaling could be simultaneously affected by *Ccn1* deletion in a single cell near the border (shown in new Supplementary Fig. S4B). In addition, being consistent with Tian et. al., cells expressing Sox9 in the TA zone lacked NICD expression, indicative of secretory progenitor cells. By contrast, most cells in the crypt bottom still exhibited a modest level of Sox9 and NICD, indicating that there might be a positional difference in whether Notch and Wnt signaling would be engaged individually or simultaneously in ISCs of the crypts. We speculate that these regulations may occur stochastically, but rather dynamically, so that different populations of cells may respond differently at a given time, making it hard to determine the exact ratio in the population. This new information is now added to the result section (p10 and Supplementary Fig. S4).

3. Minor comments are 1) In Fig. 1G, the labeling of the x-axis and the point of the graph do not match. 2) the results of Fig 4E and F are missing in the text.

→Thanks for pointing out these mistakes, which are now corrected.

Reviewer #3 (Remarks to the Author):

Comments to Authors

The authors identified that CCN1, a matricellular protein, restricts *Lgr5*+ ISCs for intestinal homeostasis via controlling Notch and Wnt signaling, which unveiled the crucial roles of matrix signaling in intestinal homeostasis and possibly intestinal regeneration. Overall, the experiments were well designed and performed. The results support the authors' statement. The statistical analyses are appropriate.

→We appreciate the reviewer's encouraging comments.

Major comments

- The authors' previous study on CCN1 in intestinal regeneration (colitis) via IL6 (Cho et al., 2015) dampens the novelty of this study.

→The previous study, as the reviewer mentioned, describes CCN1 function in mucosal repair during chronic DSS-induced colitis. However, we used *Ccn1^{dm/dm}* knock-in mice (expressing a CCN1 mutant unable to bind integrins $\alpha_6\beta_1/\alpha_M\beta_2$) as CCN1 functions on fibroblasts through integrin $\alpha_6\beta_1$ and leukocytes through $\alpha_M\beta_2$. The epithelial function of CCN1, especially on stem cells during homeostasis (using *Lgr5*+ISC-specific deletion), through integrins $\alpha_V\beta_3/\alpha_V\beta_5$ is novel and has not been examined before. We make sure this information is clearly stated in the manuscript (p20).

- It is recommended to include quantitative analysis of *Ccn1*-EGFP positive cells, villus length, and crypt depth of duodenum, jejunum, and ileum, and to briefly discuss why *Ccn1* cKO-induced phenotype was only observed in the jejunum - maybe, due to the relatively higher expression of *Ccn1* in the jejunum? Similarly, the expression pattern of integrins in the small intestine needs to be included.

→ We appreciate the reviewer for pointing this out. The explanation was not enough, concerning the use of jejunum to evaluate the expression and function of *Ccn1* in the crypt. The reason why we mainly used jejunum is to better monitor the differentiation, as some secretory cells, such as Paneth cells, are not present in the colon and a decrease in Alpi+ enterocytes toward the distal small intestine makes it difficult to obtain good alkaline phosphatase staining in the ileum (Tetteh, P. W., *et al.* 2016, Replacement of Lost Lgr5-Positive Stem Cells through Plasticity of Their Enterocyte-Lineage Daughters. *Cell Stem Cell* **18**, 203-213). As the histological evaluation showed, CCN1 expression (GFP as a surrogate) in *Ccn1^{EGFP}* mice is well observed in the crypt base of the duodenum, jejunum, ileum, and colon (newly added to Supplementary Fig. S1A-C). Similarly, the expression of integrin α_v is observed in the crypt and lamina propria. As recommended, the villus length and crypt depth were measured in the duodenum and ileum of *Ccn1^{ΔLgr5}* mice with corn oil or tamoxifen and the results are included in Supplementary Fig. S1D-G. Accordingly, we revised the result section (p6, p7, and p8)

Minor comments

- (Discussion) "We may anticipate a similar phenotype in mice with *Ccn1* deletion or mutation (D125A) upon DSS injury". The authors' previous study (2015) already showed the IBD-related phenotype in *Ccn1* ablated mice. Clarification and revision will help.

→ As explained above, we revised to clarify the use of different integrin-binding mutant mice (p20).

- It is recommended to tone down for 'knowledge gap' statement. Quite a few studies reported the importance of niche and non-cell-autonomous factors orchestrating intestinal homeostasis as well as regeneration - mainly using murine small intestine and organoids as model systems.

→ Thanks for pointing it out. We are also aware of a few non-cell autonomous factors being reported to play a role in intestinal homeostasis. In the manuscript, we tried to emphasize the fact that matrix-driven CCN1 signaling through integrins $\alpha_v\beta_3/\alpha_v\beta_5$ coordinately regulate both Wnt and Notch signaling, thus affecting the proliferation and differentiation of *Lgr5+* ISCs. As recommended, we revised such a statement in the discussion (p17).

REVIEWER COMMENTS

Reviewer #1 (Remarks to the Author):

The reviewers have not really been responsive

1. To my previous comment: "It is perhaps useful to formally show and emphasize that in this system YAP actually regulates CCN1 expression"

They responded that: "YAP regulates CCN1 expression (Fig. 6):

Unfortunately, Fig 6 does not show that. I do not see a CHIP assay, I see no reporter assays with point mutations in putative YAP binding sites in the CCN1 promoter, nor do I see any overexpression assays with CMV-YAP. All I see is an in vivo experiment showing less CCN1 expression in YAP KO mice.

2. " Recently published papers showing CCNs act in a YAP loop could also be discussed. (We are also aware that there are a couple of recently published papers regarding the regulatory loop between CCN1 and YAP. We added these to the discussion with references (p20, Ref. 65 and 66)."

I did not specifically refer to "CCN1" but "CCNs"

References 65 and 66 are:

65Li, B., et al. c-Abl regulates YAP357 phosphorylation to activate endothelial atherogenic responses to disturbed flow. *J Clin Invest* 129, 1167-1179 (2019).

66. Ege, N., et al. Quantitative Analysis Reveals that Actin and Src-Family Kinases Regulate Nuclear YAP1 and Its Export. *Cell Syst* 6, 692-708.e613 (2018).

Neither of the above papers have anything to do with a YAP autocrine loop involving CCN proteins..

Papers of interest involving CCN2 include:

Dwivedi N, Tao S, Jamadar A, Sinha S, Howard C, Wallace DP, Fields TA, Leask A, Calvet JP, Rao R. Epithelial Vasopressin Type-2 Receptors Regulate Myofibroblasts by a YAP-CCN2-Dependent Mechanism in Polycystic Kidney Disease. *J Am Soc Nephrol*. 2020 Aug;31(8):1697-1710

Dwivedi N, Tao S, Jamadar A, Sinha S, Howard C, Wallace DP, Fields TA, Leask A, Calvet JP, Rao R. Epithelial Vasopressin Type-2 Receptors Regulate Myofibroblasts by a YAP-CCN2-Dependent Mechanism in Polycystic Kidney Disease. *J Am Soc Nephrol*. 2020 Aug;31(8):1697-1710

...and, involving CCN1...

Lee S, Ahad A, Luu M, Moon S, Caesar J, Cardoso WV, Grant MB, Chaqour B. CCN1-Yes-Associated Protein Feedback Loop Regulates Physiological and Pathological Angiogenesis. *Mol Cell Biol*. 2019 Aug 27;39(18):e00107-19.

3. "Several of the experiments rely on home-made CCN1. How do the authors know that the preparation is pure and, for example, free of TGFbeta contamination? Can similar results be obtained using commercially supplied rCCN1?"

... 1) We

routinely evaluate our protein preps using SDS-gel electrophoresis, which allowed us to estimate CCN1

proteins to be >90-95% pure. However, we do not exclude the possible minor impurities that are not

visible on a gel. "

This response is incomplete, as it is not indicated how purity is evaluated. For example, neither coomassie or silver stain is adequate to identify small amounts of contaminant growth factors such as TGFbeta that have activity at pg levels. This is a major issue as CCNs are "sticky" and can be easily contaminated eg Lee S, Ahad A, Luu M, Moon S, Caesar J, Cardoso WV, Grant MB, Chaqour B. CCN1-Yes-Associated Protein Feedback Loop Regulates Physiological and Pathological Angiogenesis. Mol Cell Biol. 2019 Aug 27;39(18):e00107-19.). How did the authors specifically rule out the possibility of contaminants? Was the protein affinity purified in any way?

"2) In our organoid experiments (Fig. 4A), the add-back of CCN1-WT proteins, but not CCN1-D125A mutant proteins, reverted the phenotypes in Ccn1-deleted organoids, although they both are purified in the same way, suggesting that the effect would not be from the potential contamination."

This answer is incomplete. This mutation has been shown to affect inflammation. How can the authors be certain that this mutation does not affect the ability of CCN1 to bind inflammatory cytokines?

"3) Compared to our sf9 insect cell production system, commercially supplied CCN1 proteins are from either bacterial expression systems or Fc-fusion proteins from mammalian cells. When testing them for comparison, we found that all CCN1 proteins showed no difference (slightly better response with our CCN1 preps) in the several activities tested so far (Jun and Lau, 2020, Nature Communications). For current studies, we tested again our CCN1 proteins and Fc-fusion CCN1 protein (from R&D systems) in Ccn1-deleted organoids and found they exhibit the same activity (restriction of Lgr5+ISC expansion). "

It is important for others to reproduce the data. It is therefore essential for the authors to show these results in supplemental information. It is also important to mention, in the discussion, that it is perfectly reasonable for others to use commercially available CCN1 for studies

4. references are often incomplete

Reviewer #2 (Remarks to the Author):

The authors have properly supplemented and responded to the issues raised. It can be accepted as it is.

Reviewer #1 (Remarks to the Author):

The reviewers have not really been responsive

1. To my previous comment: "It is perhaps useful to formally show and emphasize that in this system YAP actually regulates CCN1 expression"

They responded that: "YAP regulates CCN1 expression (Fig. 6):

Unfortunately, Fig 6 does not show that. I do not see a CHIP assay, I see no reporter assays with point mutations in putative YAP binding sites in the CCN1 promoter, nor do I see any overexpression assays with CMV-YAP. All I see is an in vivo experiment showing less CCN1 expression in YAP KO mice.

→ We demonstrated YAP regulation of *Ccn1* expression in the mouse intestine using three different experimental methods. First, as the reviewer suggested, we performed the CHIP-qPCR analysis of YAP binding to the *Ccn1* promoter region, for which we used the intestinal crypts isolated from control *Ccn1^{ΔLgr5}* mice with or without administration of verteporfin, an inhibitor of YAP-TEAD interaction. The YAP-TEAD binding regions of the *Ccn1* promoter and corresponding primer sets were previously reported in Mol. Cell Biol. (2019) paper by Chaour Brahim group. Consistently, we detected the YAP enrichment on the proximal TEAD binding region (-115b) of the *Ccn1* promoter in control crypts, which was markedly reduced with verteporfin administration (Supplementary Fig.S9A). Second, a subsequent qPCR analysis showed that *Ccn1* expression was greatly reduced in the crypts of mice treated with verteporfin, confirming that YAP transcriptionally activates *Ccn1* in the mouse intestine (Supplementary Fig.S9B). Third, immunohistochemistry showed the CCN1 protein expression was also reduced in the intestine with the treatment of verteporfin (Supplementary Fig. S9C, previously S9A). Together with the reduced *Ccn1* expression in *Yap^{ΔLgr5}* mice upon tamoxifen treatment (Fig. 6A, B), these results strongly indicate YAP regulation of *Ccn1* expression in the mouse intestine. The new CHIP-qPCR of YAP binding to the *Ccn1* promoter and qPCR data on *Ccn1* expression are added to Supplementary Fig.S9A and B, and we revised the result section of the manuscripts, accordingly (p15).

2. " Recently published papers showing CCNs act in a YAP loop could also be discussed.

(We are also aware that there are a couple of recently published papers regarding the regulatory loop between CCN1 and YAP. We added these to the discussion with references (p20, Ref. 65 and 66)."

I did not specifically refer to "CCN1" but "CCNs"

References 65 and 66 are: there were mis

65Li, B., et al. c-Abl regulates YAP357 phosphorylation to activate endothelial atherogenic responses to disturbed flow. J Clin Invest 129, 1167-1179 (2019).

66. Ege, N., et al. Quantitative Analysis Reveals that Actin and Src-Family Kinases Regulate Nuclear YAP1 and Its Export. Cell Syst 6, 692-708.e613 (2018).

Neither of the above papers have anything to do with a YAP autocrine loop involving CCN proteins..

→We apologize for putting the wrong reference numbers in the previous response. We carefully examined all references cited in the manuscript.

Papers of interest involving CCN2 include:

Dwivedi N, Tao S, Jamadar A, Sinha S, Howard C, Wallace DP, Fields TA, Leask A, Calvet JP, Rao R. Epithelial Vasopressin Type-2 Receptors Regulate Myofibroblasts by a YAP-CCN2-Dependent Mechanism in Polycystic Kidney Disease. *J Am Soc Nephrol*. 2020 Aug;31(8):1697-1710

Dwivedi N, Tao S, Jamadar A, Sinha S, Howard C, Wallace DP, Fields TA, Leask A, Calvet JP, Rao R. Epithelial Vasopressin Type-2 Receptors Regulate Myofibroblasts by a YAP-CCN2-Dependent Mechanism in Polycystic Kidney Disease. *J Am Soc Nephrol*. 2020 Aug;31(8):1697-1710

...and, involving CCN1...

Lee S, Ahad A, Luu M, Moon S, Caesar J, Cardoso WV, Grant MB, Chaqour B. CCN1-Yes-Associated Protein Feedback Loop Regulates Physiological and Pathological Angiogenesis. *Mol Cell Biol*. 2019 Aug 27;39(18):e00107-19.

→We revised the discussion to include the regulatory loop between CCN proteins (CCN1 and CCN2) and YAP regulation (p20).

3. "Several of the experiments rely on home-made CCN1. How do the authors know that the preparation is pure and, for example, free of TGFbeta contamination? Can similar results be obtained using commercially supplied rCCN1?"

... 1) We

routinely evaluate our protein preps using SDS-gel electrophoresis, which allowed us to estimate CCN1 proteins to be >90-95% pure. However, we do not exclude the possible minor impurities that are not visible on a gel. "

This response is incomplete, as it is not indicated how purity is evaluated. For example, neither coomassie or silver stain is adequate to identify small amounts of contaminant growth factors such as TGFbeta that have activity at pg levels. This is a major issue as CCNs are "sticky" and can be easily contaminated eg Lee S, Ahad A, Luu M, Moon S, Caesar J, Cardoso WV, Grant MB, Chaqour B. CCN1-Yes-Associated Protein Feedback Loop Regulates Physiological and Pathological Angiogenesis. *Mol Cell Biol*. 2019 Aug 27;39(18):e00107-19.). How did the authors specifically rule out the possibility of contaminants? Was the protein affinity purified in any way?

→Our CCN1 (WT and mutants) proteins are produced in baculovirus (CCN1)-infected Sf9 insect cells that are cultured in a serum-free, protein-free Sf-900™ II SFM media (ThermoFisher #10902088), upon which CCN1 proteins are purified using ion-exchange or immuno-affinity chromatography. Then, we evaluated our proteins using SDA-PAGE and estimated the proteins to be ~95% pure, as explained in the previous response, although we do not exclude the possible minor impurities that are not visible on a gel. Upon purification, we measured the levels of endotoxin using the Limulus amoebocyte lysate (LAL) method, which showed no more than 0.08 EU per µg CCN1 proteins. We further purified the protein through a polymyxin-B-agarose

column (binding capacity of 200-500 μg LPS per 1 mg) to remove any endotoxin. These steps are critical as CCN1 promotes inflammatory cytokine expression through binding and activating Toll-like receptors 2 and 4 (Jun and Lau, 2020, Nature Communications), and we are confident that CCN1 is free of endotoxin. With regard to the specific concerns about the contamination with cytokines, specifically TGF β , we performed a TGF β ELISA assay on our CCN1 proteins and confirmed that our CCN1 preps do not contain TGF β (results are below). Of note, we also make sure that all the reagents that are used to produce our proteins are clean and free of contaminations.

Another compelling argument against the possible contribution of contaminants is the fact that multiple preparations of CCN1 protein that are expressed in very different cellular sources and purified using distinct methodologies have the same activities; our CCN1 preparation (expressed in Sf9 insect cells and purified through sequential ion-exchange chromatography) and R&D Biosystems (expressed in CHO cells as an Fc-chimera and purified by protein A affinity chromatography) showed the same activity on intestinal stem cells as shown in the previous response. Although we have not used it in the current manuscript, we have also tested CCN1 protein from Novus Biologicals (expressed in *E. coli* and purified by HPLC) for its activities in bacterial pattern binding and phagocytosis induction (Jun and Lau, 2020, Nature Communications). It is highly unlikely that proteins expressed in insect cells, hamster cells, and *E. coli* and purified using different chromatographic techniques will have the same confounding contaminants.

Furthermore, the CCN1 activities in ISC proliferation and differentiation are inhibited by an integrins α_v inhibitor, demonstrating these activities are through integrins α_v , but not other receptors (Figs. 4A, B, D, 5C, and Supplementary Fig. S8B). In addition, mutant CCN1 proteins (CCN1-D125A) that have a single amino acid substitution to render the protein to be defective in binding integrins $\alpha_v\beta_3/\alpha_v\beta_5$, although produced in the same manner as WT proteins, altered these functions on intestinal stem cells, being unable to regulate ISC homeostasis as predicted (Fig. 4A) and the phenotypes in our mouse genetic models (*Ccn1* ^{Δ Lgr5} and *Ccn1*^{D125A/D125A}) strongly support that the CCN1 functions in ISCs through integrins $\alpha_v\beta_3/\alpha_v\beta_5$ (Fig. 2). The issue of CCN1 being “sticky” is from the multiple positively charged amino acids in its CT domain; CCN1-DM mutation that changes several lysine residues and an arginine to glycines in the CT domain does not affect CCN1 ability to regulate ISCs and *Ccn1*^{DM/DM} knock-in mice showed no phenotypes (Supplementary Fig. S5C). We also previously showed that CCN1 mutant lacking its CT domain could also induce phagocytic removal of apoptotic neutrophils (Jun and Lau, 2015, Nature Communications), which requires its binding to integrins $\alpha_v\beta_3/\alpha_v\beta_5$. Mutant protein studies clearly demonstrate that these activities are attributed to the CCN1 polypeptide, but not to unknown contaminants.

As mentioned above, we have tested the reproducibility of our results using CCN1 protein purchased from commercially available sources (CCN1 expressed in CHO cells; R&D Biosystems, 4055-CR-050). Both were able to restore the phenotypes shown in the organoids with *Ccn1* deletion (Supplementary Fig. S5B). Thus, our findings can be replicated using commercially prepared CCN1 isolated from completely different cellular sources and with different purification methods.

"2) In our organoid experiments (Fig. 4A), the add-back of CCN1-WT proteins, but not CCN1-D125A mutant proteins, reverted the phenotypes in *Ccn1*-deleted organoids, although they both are purified in the same way, suggesting that the effect would not be from the potential contamination."

This answer is incomplete. This mutation has been shown to affect inflammation. How can the authors be certain that this mutation does not affect the ability of CCN1 to bind inflammatory cytokines?

→ CCN1 binding to any inflammatory cytokine has not been shown. Furthermore, CCN1 affects inflammation through cytokine gene expression via binding and activating TLR2/4 in a Myd88-dependent manner (Jun and Lau, 2020, Nature Communications). Therefore, CCN1-D125A mutation is still able to induce the expression of such genes, comparable to WT. The CCN1 functions affected upon CCN1 mutation to D125A in the context of inflammation and immune response are efferocytosis and bacterial phagocytosis, which require CCN1 binding to integrins $\alpha_v\beta_3/\alpha_v\beta_5$ on macrophages. As mentioned above, both CCN1-WT and D125A mutant proteins are expressed in Sf9 insect cells and purified the same way through sequential ion-exchange chromatography. Therefore, the change in their activities is not due to potential contamination, but the specific integrin binding.

Figure. TGFβ ELISA assay on CCN1 proteins (WT and D125A). TGFβ standards (*left*) and ELISA results on CCN1-WT and D125A proteins (*right*) are shown.

"3) Compared to our sf9 insect cell production system, commercially supplied CCN1 proteins are from either bacterial expression systems or Fc-fusion proteins from mammalian cells. When testing them for comparison, we found that all CCN1 proteins showed no difference (slightly better response with our CCN1 preps) in the several activities tested so far (Jun and Lau, 2020, Nature Communications). For current studies, we tested again our CCN1 proteins and Fc-fusion CCN1 protein (from R&D systems) in Ccn1-deleted organoids and found they exhibit the same activity (restriction of Lgr5+ISC expansion). "

It is important for others to reproduce the data. It is therefore essential for the authors to show these results in supplemental information. It is also important to mention, in the discussion, that it is perfectly reasonable for others to use commercially available CCN1 for studies

→ As suggested, we put these data in the supplementary Fig.S5B. Instead of putting it in the discussion, we added the information in the result section (p11) and method section (p22)

4. references are often incomplete

→References are now complete, including the ones the reviewer mentioned above.

Reviewer #2 (Remarks to the Author):

The authors have properly supplemented and responded to the issues raised. It can be accepted as it is.

→We appreciate the reviewer acknowledging our efforts.

REVIEWERS' COMMENTS

Reviewer #1 (Remarks to the Author):

The additions and clarifications are extremely helpful.

REVIEWERS' COMMENTS

Reviewer #1 (Remarks to the Author):

The additions and clarifications are extremely helpful.

→ We are pleased that the reviewer recognizes the additions and clarification that we made to be extremely helpful. We appreciate the reviewer's helpful comments and suggestions for our studies.